# Robust Sampling for Active Statistical Inference

**Puheng Li**
Department of Statistics
Stanford University
Stanford, CA 94305
puhengli@stanford.edu

**Tijana Zrnic**
Department of Statistics and Stanford Data Science
Stanford University
Stanford, CA 94305
tijana.zrnic@stanford.edu

**Emmanuel J. Candès**
Department of Statistics and Department of Mathematics
Stanford University
Stanford, CA 94305
candes@stanford.edu

## Abstract

Active statistical inference [51] is a new method for inference with AI-assisted data collection. Given a budget on the number of labeled data points that can be collected and assuming access to an AI predictive model, the basic idea is to improve estimation accuracy by prioritizing the collection of labels where the model is most uncertain. The drawback, however, is that inaccurate uncertainty estimates can make active sampling produce highly noisy results, potentially worse than those from naive uniform sampling. In this work, we present robust sampling strategies for active statistical inference. Robust sampling ensures that the resulting estimator is never worse than the estimator using uniform sampling. Furthermore, with reliable uncertainty estimates, the estimator usually outperforms standard active inference. This is achieved by optimally interpolating between uniform and active sampling, depending on the quality of the uncertainty scores, and by using ideas from robust optimization. We demonstrate the utility of the method on a series of real datasets from computational social science and survey research.

## 1 Introduction

Collecting high-quality labeled data remains a challenge in data-driven research, especially when each label is costly and time-consuming to obtain. In response, many fields have embraced machine learning as a practical solution for predicting unobserved labels, such as annotating satellite imagery in remote sensing [46] and predicting protein structures in proteomics [24]. Prediction-powered inference [1] is a methodological framework showing how to perform valid statistical inference despite the inherent biases in such predicted labels.

Active statistical inference [51] was recently introduced to further enhance inference by actively selecting which data points to label. The basic idea is to compute the model's uncertainty scores for all data points and prioritize collecting those labels for which the predictive model is most uncertain. When the uncertainty scores appropriately reflect the model's errors, Zrnic and Candès [51] show that active inference can significantly outperform prediction-powered inference (which can essentially be thought of as active inference with naive uniform sampling), meaning it results in more accurate estimates and narrower confidence intervals. However, when uncertainty scores are of poor quality, active inference can result in overly noisy estimates and large confidence intervals. This

39th Conference on Neural Information Processing Systems (NeurIPS 2025).

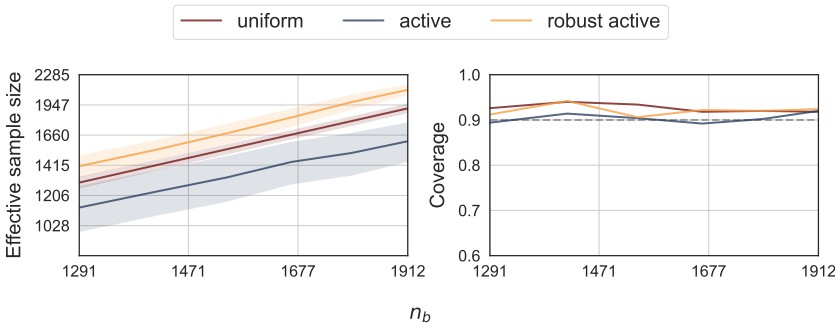

Figure 1: **Effective sample size and coverage on Pew post-election survey data.** We compare uniform, active, and robust active sampling, for different values of the sampling budget $n_b$. The target of inference is the approval rate of a presidential candidate. We show one standard deviation around the effective sample size.

is an important limitation, seeing that there is widespread recognition that measuring model uncertainty is challenging. Large language models, for example, are often overconfident in their answers [10, 47, 48]. Miscalibrated uncertainty scores also arise when there is a distribution shift between the training data and the test domain.

To illustrate the issue empirically, consider the problem of estimating the approval rate of a presidential candidate: $\theta^* = \mathbb{E}[Y]$, where $Y \in \{0, 1\}$ is the binary indicator of approval, using Pew post-election survey data [32]. Here, we have demographic covariates $X_1, \ldots, X_n$ corresponding to $n$ people, but we do not observe the approval indicator $Y_i$ for everyone. Rather, we have a budget $n_b < n$ on how many people we can survey and collect their $Y_i$. In addition, we have a machine learning model $f$ that we can use to obtain a cheap prediction $f(X_i)$ of $Y_i$ from the available covariates. Active inference suggests surveying those individuals where $f$ is uncertain. For example, if $f(X_i)$ is obtained by thresholding a continuous score $p(X_i) \in [0, 1]$ representing the probability the model assigns to the missing label taking on the value 1, this could mean prioritizing the collection of labels where $p(X_i)$ is close to $0.5$. In Figure 1, we show the effective sample size and coverage of prediction-powered inference (uniform sampling), standard active inference, and our robust active inference method, for varying values of the budget $n_b$. The effective sample size is formally defined in Section 4; it is the number of samples the method that samples uniformly at random would need to use to achieve the accuracy of the labeling method under study. To demonstrate a challenge for active inference, we train $f$ on a small dataset, resulting in poorly estimated uncertainties. We see that active sampling results in a smaller effective sample size than simple uniform sampling. This is because the variance of the active sampling strategy is large. Meanwhile, the robust method outperforms both baselines. This is achieved by estimating the quality of the uncertainty scores and optimally interpolating between uniform and active sampling. All three methods come with provable validity guarantees, as confirmed by the achieved target coverage of $90\%$.

The source code for all experiments is available at `https://github.com/lphLeo/Robust-Active-Statistical-Inference`.

## 1.1 Related work

Our paper builds on active statistical inference [51], which itself builds on prediction-powered inference [1] and, more generally, statistical inference assisted by predictive models [29, 40, 45]. There is a growing literature in this space, aimed at ensuring robustness against poor predictions [2, 16, 19, 23, 30, 31], sample efficiency when there is no good pre-trained model $f$ [52], simplicity and applicability to more general estimation problems [25, 50], and handling missing covariates [25, 31]. Notably, several works study adaptive label collection strategies [3, 14, 17].

Zooming out further, at a technical level this line of work relates to semiparametric inference, missing data, and causality [34, 35, 37, 44]. In particular, the prediction-powered and active inference estimators closely resemble the augmented inverse probability weighting (AIPW) estimator [35].

Our work also connects with many areas in machine learning and statistics that study adaptive data collection; most notably, active learning [36, 39] and adaptive experimental design [13, 21]. We collect data based on model uncertainty, akin to active learning; however, our objective is statistical inference on typically low-dimensional parameters, rather than prediction. Active testing [26] also involves adaptive data collection, but it pursues a different objective of high-precision risk estimation for a fixed model and uses a distinct estimator. Our approach can be seen as an adaptive design assisted by a powerful predictive model, with a robustness wrapper for improved performance.

More distantly, our work also relates to robust statistics and robust machine learning [8, 20, 22, 33, 41, 42, 49]. In particular, our method provides a safeguard against poor uncertainty estimation by solving a robust optimization problem [5, 6, 15].

## 1.2 Problem setup

We follow the problem setting from [51]. We observe unlabeled instances $X_1, \ldots, X_n$ drawn i.i.d. from a distribution $P_X$, but we do not observe their labels $Y_i$. We use $P = P_X \times P_{Y|X}$ to denote the joint distribution of $(X_i, Y_i)$. Our goal is to perform inference for a parameter $\theta^*$ that depends on the distribution of the unobserved labels; that is, the parameter is a functional of $P$. In particular, we assume that $\theta^*$ can be written as:

$$\theta^* = \arg\min_{\theta} \mathbb{E}\left[\ell_\theta(X, Y)\right], \text{ where } (X, Y) \sim P.$$

Here, $\ell_\theta$ is a convex loss function. This is a broad class of estimands, known as *M-estimation*, and it includes means, medians, linear and logistic regression coefficients, and more. We have a budget $n_b$ on the number of labels we can collect in expectation, and typically $n_b \ll n$. To assist in imputing the missing labels, we also have a black-box predictive model $f$ at our disposal.

## 2 Warm-up: robust sampling for mean estimation

Consider the case where $\theta^*$ is the label mean, $\theta^* = \mathbb{E}[Y]$. The active inference estimator for $\theta^*$ is given by:

$$\hat{\theta}^\pi = \frac{1}{n} \sum_{i=1}^{n} \left( f(X_i) + (Y_i - f(X_i)) \frac{\xi_i}{\pi(X_i)} \right). \tag{1}$$

Here, $\pi(\cdot)$ is any sampling rule that satisfies $\mathbb{E}[\pi(X)] \leq \frac{n_b}{n}$ so that the budget constraint is met on average, and $\xi_i \sim \text{Bern}(\pi(X_i))$ is the indicator of whether the label $Y_i$ is sampled. Since the number of labeled data points is a sum of independent Bernoullis, a standard Hoeffding argument guarantees that the realized labeling rate will closely match the budget with high probability. Specifically, the labeling ratio will not exceed $\frac{n_b}{n} + \epsilon$ with probability $1 - \delta$, provided that $n > \frac{\log(1/\delta)}{2\epsilon^2}$ for any $\epsilon, \delta > 0$. As shown in [51], the variance of this estimator is

$$\text{Var}\left(\hat{\theta}^\pi\right) = \frac{1}{n}\left(\text{Var}(Y) + \mathbb{E}\left[(Y - f(X))^2\left(\frac{1}{\pi(X)} - 1\right)\right]\right), \tag{2}$$

and the optimal sampling rule is $\pi_{\text{opt}}(X_i) \propto \sqrt{\mathbb{E}[(Y_i - f(X_i))^2 | X_i]}$. In other words, it is optimal to upsample where the model $f$ makes the largest errors.

The most straightforward sampling rule that satisfies the budget constraint is the uniform rule: $\pi^{\text{unif}}(X) = n_b/n$. However, if we have access to a good measure of model uncertainty that can serve as a proxy for the model error $\sqrt{\mathbb{E}[(Y_i - f(X_i))^2 | X_i]}$, then we can obtain a rule that is closer to $\pi_{\text{opt}}$. For example, we might prompt a large language model for its uncertainty about $X_i$ or look at the softmax output of a neural network, and upsample where the uncertainty is high. The issue is that if we severely underestimate the model error, then the estimator's variance can blow up: clearly, if $\pi(X_i)$ is small when the actual error $(Y_i - f(X_i))^2$ is large, the variance will be large as well. This is the reason why we saw poor performance in Figure 1.

Given any initial sampling rule $\pi$, our approach is to find an improved, *robust* sampling rule $\pi^{\text{robust}}$ that is never worse than either $\pi$ or $\pi^{\text{unif}}$. By that we mean that the resulting active inference estimator will have a variance that is no worse that with either $\pi$ or $\pi^{\text{unif}}$ used for label collection: $\text{Var}(\hat{\theta}^{\pi^{\text{robust}}}) \leq \min\{\text{Var}(\hat{\theta}^\pi), \text{Var}(\hat{\theta}^{\pi^{\text{unif}}})\}$.

## 2.1 Budget-preserving path

Since our goal is to find a sampling rule $\pi^{\text{robust}}$ that performs no worse than $\pi^{\text{unif}}$ and an arbitrary given $\pi$, it is natural to consider a path that connects $\pi$ and $\pi^{\text{unif}}$, while preserving the sampling budget along the path.

**Definition 1** (Budget-preserving path). We call a continuous path $\pi^{(\rho)}$, $\rho \in [0, 1]$, a *budget-preserving path* connecting $\pi$ and $\pi^{\text{unif}}$ if $\pi^{(0)} = \pi$, $\pi^{(1)} = \pi^{\text{unif}}$, and $\mathbb{E}[\pi^{(\rho)}(X)] = \mathbb{E}[\pi(X)]$ for all $\rho \in [0, 1]$.

Correspondingly, given a point $\rho$ along the path, we compute the estimator $\hat{\theta}^{\pi^{(\rho)}}$, obtained as the active inference estimator (1) with sampling rule $\pi^{(\rho)}$. The following are some examples of valid budget-preserving paths.

**Example 1** (Linear path). $\pi^{(\rho)} = (1 - \rho)\pi + \rho\pi^{\text{unif}}$.

**Example 2** (Geometric path). $\pi^{(\rho)} \propto \pi^{1-\rho}(\pi^{\text{unif}})^{\rho}$. The "$\propto$" hides the normalization factor that ensures $\mathbb{E}[\pi^{(\rho)}(X)] = \mathbb{E}[\pi(X)]$ for all $\rho$.

A natural family of budget-preserving paths can be recovered via the "least-action" principle, yielding the definition of geodesic paths. See Appendix B for a general definition of geodesic paths, details of how Examples 1 and 2 can be recovered as special cases, as well as further examples.

Of course, if we consistently estimate the optimal point $\rho^* \in [0, 1]$ along the path, we are guaranteed to find an estimator that outperforms naive active inference and uniform sampling. Moreover, the resulting estimator is still asymptotically normal, which permits the construction of valid confidence intervals. We formalize this key result below in which $\sigma_\rho^2 = n\text{Var}(\hat{\theta}^{\pi^{(\rho)}})$.

**Theorem 1.** Suppose $\pi^{(\rho)}$ is a budget-preserving path connecting $\pi$ and $\pi^{\text{unif}}$. Let $\rho^* = \arg\min_\rho \text{Var}(\hat{\theta}^{\pi^{(\rho)}})$, and suppose $\hat{\rho} = \rho^* + o_P(1)$. Then,

$$\sqrt{n}\left(\hat{\theta}^{\pi^{(\hat{\rho})}} - \theta^*\right) \xrightarrow{d} \mathcal{N}\left(0, \sigma_{\rho^*}^2\right),$$

where $\sigma_{\rho^*}^2 \leq \min\{\sigma_0^2, \sigma_1^2\}$.

Theorem 1 shows that consistently estimating $\rho^*$ will result in an estimator that is no worse than either endpoint. If $\rho^*$ is additionally unique and within $(0, 1)$, then the resulting sampling will strictly outperform both active sampling with $\pi$ and uniform sampling. The theoretical results in this paper are asymptotic; however, validity in the finite-sample regime is shown empirically in the experiments in Section 4.

It remains to explain how to estimate $\hat{\rho}$. Recall from (2) that $\text{Var}(\hat{\theta}^{\pi^{(\rho)}}) = \frac{1}{n}\mathbb{E}[\frac{e^2(X)}{\pi^{(\rho)}(X)}] + C$, where $e^2(X) = \mathbb{E}[(Y - f(X))^2|X]$ and $C$ is a quantity that has no dependence on $\pi^{(\rho)}$. Therefore, to fit $\hat{\rho}$, we fit an error function $\hat{e}^2(\cdot) \approx e^2(\cdot)$ and solve for the $\rho$ that minimizes the empirical approximation of $\text{Var}(\hat{\theta}^{\pi^{(\rho)}})$:

$$\hat{\rho} = \arg\min_\rho \frac{1}{n}\sum_{i=1}^n \frac{\hat{e}^2(X_i)}{\pi^{(\rho)}(X_i)}. \tag{3}$$

We can find the solution by performing a grid search over $\rho \in [0, 1]$. The error $\hat{e}^2(\cdot)$ can be fit on historical or held-out data, or it can be gradually fine-tuned during the data collection process. Notice that, if the error estimation is consistent in the sense that $\|\hat{e}^2(X) - e^2(X)\|_\infty \xrightarrow{p} 0$ and if $\rho^*$ is unique, then $\hat{\rho} \xrightarrow{p} \rho^*$, as assumed in Theorem 1. Here, the assumption that $\hat{e}$ converges to $e$ follows from classical arguments of uniform approximation of flexible estimators, and is common in the field of semiparametric inference. For instance, the widely-used doubly robust estimator [18, 35], which is closely related to our estimator, relies on consistent estimation of nuisance functions.

## 2.2 Robustness to error function misspecification

Given a path $\pi^{(\rho)}$, the previous discussion suggests finding $\hat{\rho}$ that minimizes an empirical approximation of the variance $\text{Var}(\hat{\theta}^{\pi^{(\rho)}})$. This empirical approximation relies on an error estimate $\hat{e}(\cdot)$. If

this function is severely misspecified, then the computed $\hat{\rho}$ might be far from $\rho^*$; more importantly, it might not even outperform uniform sampling.

To mitigate this concern, we instead consider a robust optimization problem that incorporates the possibility of $\hat{e}$ being misspecified:

$$\rho_{\text{robust}} = \arg\min_{\rho} \max_{\boldsymbol{\epsilon} \in \mathcal{C}} \frac{1}{n} \sum_{i=1}^{n} \frac{\hat{e}^2(X_i) + \epsilon_i}{\pi^{(\rho)}(X_i)}. \tag{4}$$

Here, $\boldsymbol{\epsilon} = (\epsilon_1, \ldots, \epsilon_n)$ is the misspecification vector and $\mathcal{C}$ is the admissible set of misspecifications. This method allows for setting $\pi^{(\rho)}$ close to uniform if the misspecification set $\mathcal{C}$ is permissive enough. Solving this minimax problem is computationally efficient, as long as $\mathcal{C}$ is a convex set. The outer problem can be solved via a one-dimensional grid search, while the inner problem is tractable due to convexity.

Now, the question is how we should set $\mathcal{C}$ in practice. Our default will be to simply use $\mathcal{C} = \{\boldsymbol{\epsilon} : \|\boldsymbol{\epsilon}\|_2 \leq c\}$, for some hyperparameter $c > 0$. Empirically, $c$ can be set by cross-validation. Other choices of the set $\mathcal{C}$ are possible, such as bounding other norms of $\boldsymbol{\epsilon}$, for example $\|\boldsymbol{\epsilon}\|_1 < c$. Empirically we found the $\ell_2$ norm to work the best, and in illustrative theoretical examples we reach the same conclusion; see Appendix C for details. We also tried relative misspecification, in the sense that $\epsilon_i = \hat{e}^2(X_i)(1 + \eta_i)$, and constrained either the $\ell_1$ or $\ell_2$ norm of the relative perturbation $\eta$. We found that this does not perform as well.

Zrnic and Candès [51] briefly discussed a robustness proposal with linear interpolation. It assumes access to historical data, and otherwise it selects a default value for the coefficient, which has no guarantee to outperform uniform and active sampling. Our analysis is far more thorough and systematic, expanding the set of interpolating paths, not requiring historical data but incorporating a burn-in period, and adding a robustness constraint. These are all crucial for the practicality and reliability of the method; see Section 4 for details.

There are other potential optimization objectives to take into account robustness constraints. For example, one may penalize small values of $\rho$ in the objective (3) with regularization, and similarly use cross-validation to choose the penalty parameter. We leave the investigation of such alternatives for future work.

## 3 Robust sampling for general M-estimation

Our sampling principle can be directly extended to general convex M-estimation, as considered in [51]. We explain this step-by-step for completeness.

Recall that we consider all inferential targets of the form $\theta^* = \arg\min_{\theta} \mathbb{E}\left[\ell_\theta(X, Y)\right]$, for a convex loss $\ell_\theta$. Denote $\ell_{\theta,i} = \ell_\theta(X_i, Y_i)$, $\ell_{\theta,i}^f = \ell_\theta(X_i, f(X_i))$, and define $\nabla \ell_{\theta,i}$ and $\nabla \ell_{\theta,i}^f$ similarly. For an active sampling strategy $\pi$, the general active inference estimator is defined as:

$$\hat{\theta}^\pi = \arg\min_{\theta} L^\pi(\theta), \text{ where } L^\pi(\theta) = \frac{1}{n} \sum_{i=1}^{n} \left(\ell_{\theta,i}^f + \left(\ell_{\theta,i} - \ell_{\theta,i}^f\right) \frac{\xi_i}{\pi(X_i)}\right). \tag{5}$$

As before, $\xi_i \sim \text{Bern}(\pi(X_i))$ is the indicator of whether the label $Y_i$ is sampled. Following [51], we know that the asymptotic covariance matrix of $\hat{\theta}^\pi$ equals:

$$\Sigma^\pi = H_{\theta^*}^{-1} \text{Var}\left(\nabla \ell_{\theta^*}^f + \left(\nabla \ell_{\theta^*} - \nabla \ell_{\theta^*}^f\right) \frac{\xi}{\pi(X)}\right) H_{\theta^*}^{-1},$$

where $H_{\theta^*}$ is the Hessian $H_{\theta^*} = \nabla^2 \mathbb{E}\left[\ell_{\theta^*}(X, Y)\right]$.

We again consider budget-preserving paths $\pi^{(\rho)}$ and tune the parameter $\rho$ such that we minimize the variance of the resulting estimator $\hat{\theta}^{\pi^{(\rho)}}$. Denote by $\Sigma_0$ and $\Sigma_1$ the asymptotic covariance matrices of the active inference estimator (5) using $\pi^{(0)} = \pi$ and $\pi^{(1)} = \pi^{\text{unif}}$, respectively.

**Theorem 2.** Suppose $\pi^{(\rho)}$ is a budget-preserving path connecting $\pi$ and $\pi^{\text{unif}}$. Given a coordinate $j$ of interest, let $\rho^* = \arg\min_{\rho} \Sigma_{jj}^{\pi^{(\rho)}}$, and suppose $\hat{\rho} = \rho^* + o_P(1)$. Suppose further that $\hat{\theta}^{\pi^{(\rho^*)}} \xrightarrow{p} \theta^*$.

Then,

$$\sqrt{n}\left(\hat{\theta}^{\pi^{(\hat{\rho})}} - \theta^*\right) \xrightarrow{d} \mathcal{N}\left(0, \Sigma_{\rho^*}\right),$$

where $\Sigma_{\rho^*, jj} \leq \min\{\Sigma_{0,jj}, \Sigma_{1,jj}\}$.

The consistency condition $\hat{\theta}^{\pi^{(\rho^*)}} \xrightarrow{p} \theta^*$ is standard; see the corresponding discussion in [51] and [2]. For example, it is ensured when $L^\pi$ is convex, such as in the case of generalized linear models (GLMs), or when the parameter space is compact.

As in the case of mean estimation, we fit $\hat{\rho}$ by approximating the variance of the estimator $\Sigma^{\pi^{(\rho)}}$ and searching over $\rho$. However, here the notion of error $e^2(\cdot)$ we need to estimate is different. In particular, given the form of $\Sigma^\pi$, we let

$$\hat{\rho} = \arg\min_\rho \frac{1}{n} \sum_{i=1}^n \frac{\hat{e}^2(X_i)}{\pi^{(\rho)}(X_i)},$$

where $\hat{e}^2(X)$ aims to approximate $e^2(X) = \mathbb{E}[((\nabla\ell_{\theta^*} - \nabla\ell_{\theta^*}^f)^\top h^{(j)})^2 | X]$ and $h^{(j)}$ is the $j$-th column of $H_{\theta^*}^{-1}$. In the context of generalized linear models (GLMs), this error simplifies to $e^2(X) = \mathbb{E}[(Y - f(X))^2 | X] \cdot (X^\top h^{(j)})^2$. Therefore, as for mean estimation, the problem essentially reduces to estimating the error $\mathbb{E}[(Y - f(X))^2 | X]$. As before, if $\hat{e}^2$ consistently estimates $e^2$, then $\hat{\rho}$ consistently estimates $\rho^*$.

Finally, to protect against poorly estimated errors $\hat{e}$, we can incorporate an uncertainty set $\mathcal{C}$ around the error estimates just as before (4). Again, the only difference here is that the $\hat{e}^2(X_i)$'s are estimating a different notion of model error tailored to the inference problem at hand.

We summarize our general *robust active inference* algorithm in Algorithm 1.

---

**Algorithm 1:** Robust Active Inference

**Input:** unlabeled data $X_1, \ldots, X_n$, labeling budget $n_b$, predictive model $f$, initial sampling rule $\pi$, budget-preserving path $\pi^{(\rho)}$, error estimator $\hat{e}^2(\cdot)$, robustness constraint $\mathcal{C}$

1 Solve the minimax problem $\rho_{\text{robust}} = \arg\min_{\rho \in [0,1]} \max_{\epsilon \in \mathcal{C}} \frac{1}{n} \sum_{i=1}^n \frac{\hat{e}^2(X_i) + \epsilon_i}{\pi^{(\rho)}(X_i)}$

2 Sample labeling decisions according to $\pi^{(\rho_{\text{robust}})}(X_i)$: $\xi_i \sim \text{Bern}\left(\pi^{(\rho_{\text{robust}})}(X_i)\right), i \in [n]$

3 Collect labels $\{Y_i : \xi_i = 1\}$

**Output:** estimator $\hat{\theta}^{\pi^{(\rho_{\text{robust}})}} = \arg\min_\theta L^{\pi^{(\rho_{\text{robust}})}}$, as defined in Eq. (5)

---

## 4 Experiments

We turn to evaluating the performance of our robust sampling approach empirically. Each of the following subsections is dedicated to a different experiment using social science research data. Section 4.1 measures presidential approval, Section 4.2 analyzes US age–income patterns, and Section 4.3 applies language models to score text on social attributes such as political bias. On each of these datasets, we use the following methods to collect labels: (1) uniform sampling, which essentially recovers prediction-powered inference [1]; (2) standard uncertainty-based active sampling [51]; and (3) our robust active method as per Algorithm 1. Each dataset will use a different base predictive model $f$, which we describe therein. We set the target coverage level to be 0.9 throughout.

The main metric used for the comparison is effective sample size. To define this metric formally, consider the baseline estimator that samples uniformly at random, i.e., according to $\pi^{\text{unif}}$. Its effective sample size is simply its budget $n_b$. For other estimators, we say that the effective sample size is equal to $n_{\text{eff}}$ if the estimator achieves the same variance as the baseline estimator with budget $n_{\text{eff}}$. For example, if given budget $n_b = 100$ the estimator achieves the same variance as the baseline estimator with double the budget, then the estimator has $n_{\text{eff}} = 200$. A larger $n_{\text{eff}}$ indicates a more efficient estimator. In the case where the effective sample size falls below the budget, $n_{\text{eff}} < n_b$, the estimator performs worse than the baseline. We show one standard deviation around the effective sample size in all plots.

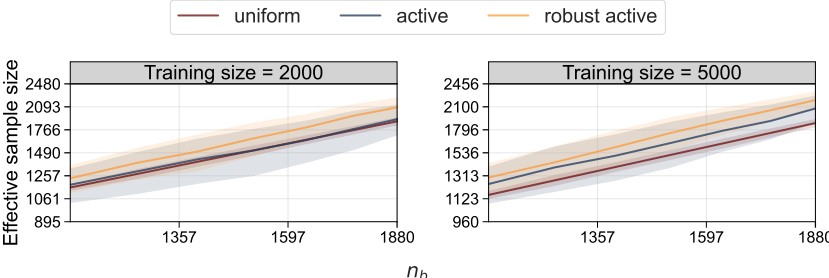

Figure 2: **Effective sample size on Pew post-election survey data**, for different dataset sizes used to train $f$. We compare uniform, active, and robust active sampling, for different values of the sampling budget $n_b$. The target of inference is the approval rate of a presidential candidate.

We also plot empirical estimates of the methods' coverage. We estimate the coverage by resampling the data, constructing confidence intervals for each resampling, and calculating the proportion of times the true parameter value (approximated by the full-data estimate of the target $\theta^*$) falls within the constructed intervals. This approach allows us to assess how reliably each method achieves the target coverage level. We resample 500 times to estimate the coverage. (We note that this approach yields conservative coverage estimates when $n_b$ is large, because we have $n - n_b$ "fresh" labels to approximate $\theta^*$.) From the theory, we know that the coverage should be exactly 0.9 for all baselines.

### 4.1   Post-election survey research

Following [51], we evaluate the different methods on survey data collected by the Pew Research Center following the 2020 United States presidential election, aiming at gauging people's approval of the presidential candidates' political messaging [32]. We aim to estimate the approval rate $\theta^* = \mathbb{E}[Y]$, where $Y \in \{0,1\}$ is a binary indicator of approval of Biden's political messaging. We use a multilayer perceptron (MLP) as our predictive model $f$. At the beginning, we have a "burn-in" period where we collect all burn-in labels $Y_i$ and we use this burn-in data to estimate the error function $\hat{e}(\cdot)$. Afterwards, we use the fitted function to run robust active inference, as per Algorithm 1. Naturally, the burn-in period counts towards the overall labeling budget $n_b$.

We study three questions: (1) the effect of tuning $\rho$ along the budget-preserving path, without incorporating a robustness constraint $\mathcal{C}$; (2) the effect of tuning $\rho$ along the path and the robust optimization over $\mathcal{C}$ combined; and (3) the performance of different budget-preserving paths.

**Tuning along the budget-preserving path.**   First, we conduct an experiment without the robustness set $\mathcal{C}$, only tuning the parameter $\hat{\rho}$ along the budget-preserving path. We choose the geometric path from Example 2. To implement active inference, we use $\pi(x) \propto \min\{f(x), 1 - f(x)\}$, in which $f(x)$ is the predicted probability that the label takes on the value 1, as considered in [51]. See Figure 2 for the results. We consider two training dataset sizes used to train $f$, allowing us to see the results for a less accurate $f$ (left) and a more accurate one (right). We find that, even without robust optimization but only optimizing along the budget-preserving path, robust active inference can lead to noticeable improvements in terms of power compared to naive uncertainty-based active sampling and uniform sampling. The performance of standard active inference crucially depends on the quality of $f$ and its uncertainties. We defer the corresponding coverage plots to Appendix E.

**Incorporating robustness.**   One strategy proposed by Zrnic and Candès [51] is to estimate $\hat{e}$ and set $\pi$ proportional to $\hat{e}$. With this choice, without the additional step of robust optimization, our robust sampling approach would trivially estimate $\hat{\rho} = 0$ (a proof of this claim can be found in Appendix A). We show that incorporating the robustness constraint resolves this issue when $\pi(x) \propto \hat{e}(x)$. As in the previous case, we use the geometric path and an MLP as the predictive model. The results are shown in Figure 3. Recall, $\hat{e}$ is estimated from the burn-in data. Thus, the longer the burn-in period, the better the fit $\hat{e}$. This is consistent with the observation that active inference gradually outperforms uniform sampling as the burn-in period grows. However, when there is little data to fit $\hat{e}$, active sampling leads to a significantly higher variance than uniform sampling. Our robust sampling approach is never worse than either baseline, across all burn-in data sizes. This is explained by the

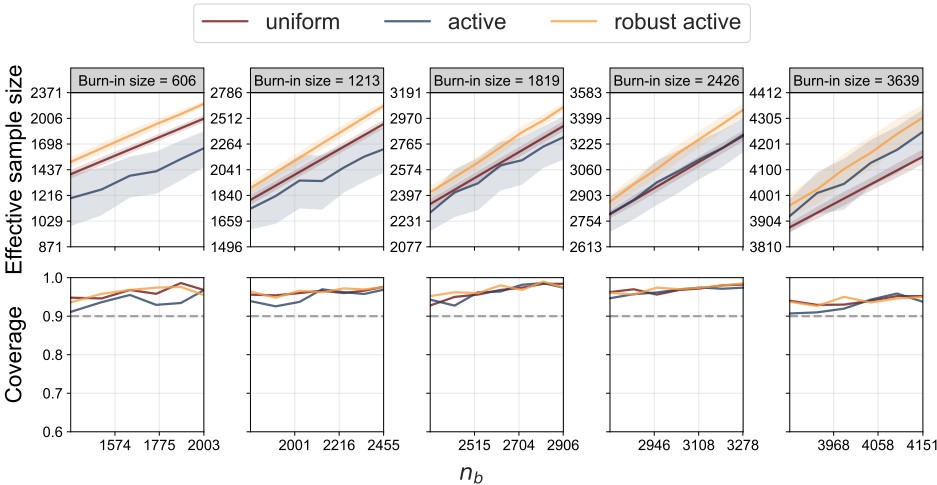

Figure 3: **Effective sample size (top) and coverage (bottom) on Pew post-election survey data**, for varying burn-in dataset sizes with respect to different proportions of the data. We compare uniform, active, and robust active sampling, for different values of the sampling budget $n_b$. The target of inference is the approval rate of a presidential candidate.

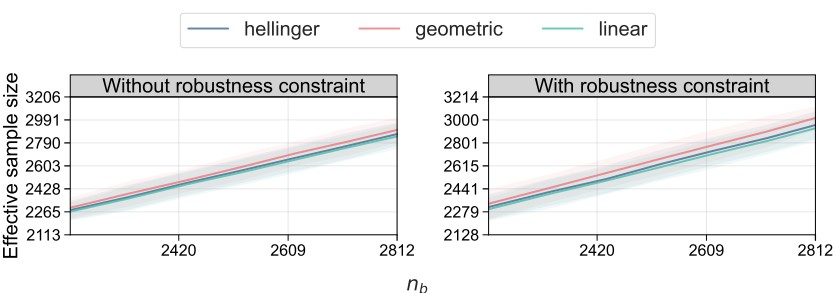

Figure 4: **Effective sample size for different budget-preserving paths on Pew post-election survey data**, without (left) and with (right) a robustness constraint $\mathcal{C}$. In both cases, the geometric path leads to the largest effective sample size. The target of inference is the same as in Figure 3.

fact that, when the fit $\hat{e}$ is poor, the constraint set $\mathcal{C}$ chosen via cross-validation is large, resulting in a large $\rho_{\mathrm{robust}}$, thus pushing the sampling rule closer to uniform. In Figure 6 (left), we plot the optimized value $\rho_{\mathrm{robust}}$ for different burn-in sizes. As expected, $\rho_{\mathrm{robust}}$ decreases, which means that the optimal strategy gradually moves from uniform sampling toward standard active sampling as the quality of $\hat{e}$ improves.

**Choice of budget-preserving path.**  We have thus far used the geometric path as our budget-preserving path. In Figure 4 we compare three budget-preserving paths: the linear path, the geometric path, and the Hellinger path (see Appendix B). On the post-election survey dataset, Figure 4 shows that the geometric path is the best of the three chosen paths, regardless of whether or not robust optimization over $\mathcal{C}$ is used. Therefore, as a practical default, we recommend using the geometric path. It has been stress-tested and has consistently demonstrated strong performance in our evaluations. We believe this is a good tradeoff between simplicity and performance. For improved performance with a better choice of path, the practitioner might want to tune it in a data-driven way; for example, based on the estimated variance on a small held-out dataset.

## 4.2 Census data analysis

We study the annual American Community Survey (ACS) Public Use Microdata Sample (PUMS) collected by the US Census Bureau [12]. We are interested in investigating the relationship between

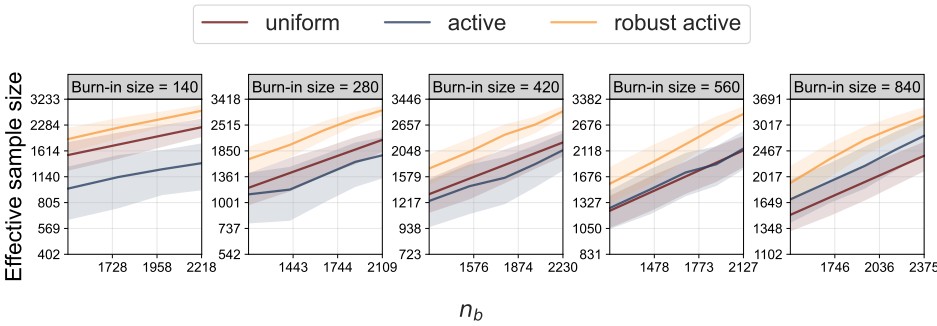

Figure 5: **Effective sample size on US Census data**, for varying burn-in dataset sizes. We compare uniform, active, and robust active sampling, for different values of the sampling budget $n_b$. The target of inference is the relationship between age and income, estimated via a linear regression.

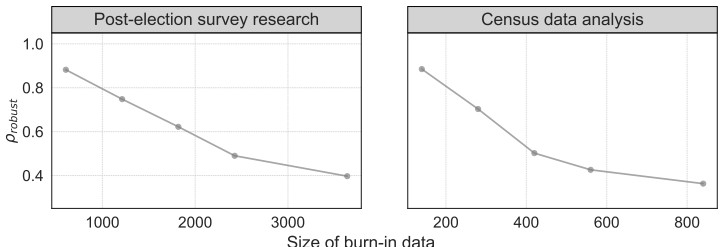

Figure 6: **Optimized value $\rho_{\mathrm{robust}}$ along the geometric path** as a function of the size of the burn-in data for the post-election survey data (left) and US Census data (right).

age and income in survey data collected in California in 2019, controlling for sex. We estimate the age coefficient $\theta^*$ of the linear regression vector when regressing income on age and sex. We use an XGBoost model [9] to predict income $Y$ from available demographic covariates. As in the previous problem, we set $\pi(x) \propto \hat{e}(x)$, as in [51], and fit $\hat{e}$ using burn-in data. We use the geometric path and incorporate the robust optimization over $\mathcal{C}$. We show the results in Figure 5. Again, we observe that the robust approach outperforms both standard active sampling and uniform sampling for different qualities of the error estimate $\hat{e}(\cdot)$, corresponding to different burn-in dataset sizes. Standard active inference, on the other hand, is very sensitive to the quality of $\hat{e}$. In Figure 6 (right), we plot the optimized value $\rho_{\mathrm{robust}}$ for different burn-in sizes. As in the previous example, $\rho_{\mathrm{robust}}$ decreases as the quality of $\hat{e}$ improves, as expected. We include corresponding coverage plots in Appendix E.

### 4.3 Computational social science with language models

We study three text annotation tasks used for computational social science research. In each task, we have text instances $X_i$ and we seek to collect labels $Y_i$ related to the text's sentiment, political leaning, and so on. We wish to use a large language model (LLM) $f$ to predict the high-quality annotations $Y_i$, which are typically collected through laborious human annotation. A natural way of actively sampling human annotations is according to the confidence of the language model [17, 27]. Tian et al. [43] propose prompting LLMs to verbalize their confidence in the provided answer, and they find that this results in fairly calibrated confidence scores. Gligorić et al. [17] find that such scores can be useful in actively sampling human annotations. We use GPT-4o annotations and confidences collected by Gligorić et al. [17]. We apply active inference with $\pi(X_i) \propto (1 - C_i)$, where $C_i$ is the collected confidence score of the language model for prompt $X_i$. This can be a brittle strategy, since the scores are often overconfident and thus result in very small sampling probabilities, which can blow up the estimator variance through inverse probability weighting. For robust active inference, we use the geometric path and robust optimization with an $\ell_2$ constraint set $\mathcal{C}$, as before.

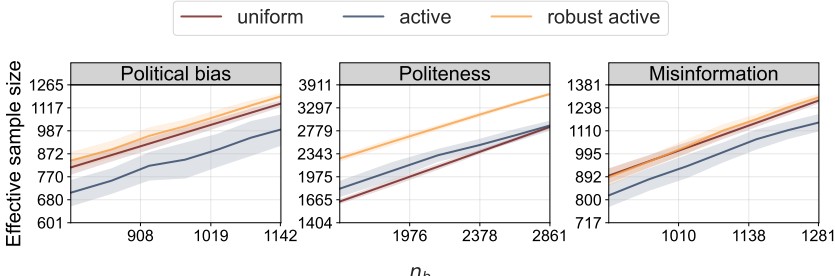

Figure 7: **Effective sample size on social science text annotation datasets**. We compare uniform, active, and robust active sampling, for different values of the sampling budget $n_b$. The targets of inference are (left to right) the prevalence of right-leaning political bias, the relationship between hedging and politeness, and the prevalence of misinformation.

**Political bias.**    In the first task, the goal is to study the political leaning of media articles, using the data curated by Baly et al. [4]. The labels $Y$ are one of `left`, `centrist`, or `right`. The inferential target is the prevalence of right-leaning articles: $\theta^* = \mathbb{E}[\mathbf{1}\{Y = \texttt{right}\}]$.

**Politeness.**    The next task is to estimate how certain linguistic devices impact the perceived politeness of online requests. We use the dataset of requests from Wikipedia and StackExchange curated by Danescu-Niculescu-Mizil et al. [11]. We study how the presence of hedging in the request, $X_{\text{hedge}} \in \{0, 1\}$, impacts whether a text is seen as polite, $Y \in \{0, 1\}$. Formally, $\theta^*$ is this effect estimated via a logistic regression with an intercept: $\text{logit}\left(\mathbb{P}\left(Y = 1 \mid X_{\text{hedge}}\right)\right) = \theta_0 + \theta^* X_{\text{hedge}}$.

**Misinformation.**    Finally, we study the prevalence of misinformation in news headlines, using the dataset collected by Gabriel et al. [15]. The labels $Y \in \{0, 1\}$ indicate whether a headline contains misinformation. The inferential target is the prevalence of misinformation, $\theta^* = \mathbb{E}[Y]$.

We show the results in Figure 7. Across all tasks, the robust approach is essentially never worse than uniform sampling or active inference, in cases even outperforming both by a large margin. Standard active inference often leads to large intervals, given that sampling directly according to the model's verbalized uncertainty leads to instability through inverse probability weighting. We include the corresponding coverage plots in Appendix E.

## 5   Conclusion

We presented robust sampling strategies for active inference: a principled hedge between uniform and conventional active sampling. By selecting an optimal tuning parameter $\rho$ along a budget-preserving path, robust active inference ensures performance that is no worse than with standard active sampling, and it reduces to near-uniform sampling when uncertainty scores are unreliable. Furthermore, the estimator can even surpass standard active inference given reliable uncertainties.

Many directions remain for future work. For example, it would be valuable to understand how to optimally choose the constraint set $\mathcal{C}$, or at least how to choose between several different constraint sets. As presented, our procedure is sensitive to the choice of $\mathcal{C}$ and may result in sampling rules that are too close or too far from uniform if this set is chosen poorly. We also leave investigations into the optimal budget-preserving path, and practical heuristics for how a practitioner might effectively choose a good path in a data-driven way, for future work.

## Acknowledgement

EJC was supported by the Office of Naval Research grant N00014-24-1-2305, the National Science Foundation grant DMS-2032014, and the Simons Foundation under award 814641.

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

# A  Proofs

## A.1  Proof of Theorem 1

By the definition of $\hat{\theta}^{\pi^{(\rho)}}$, we have

$$\hat{\theta}^{\pi^{(\rho)}} = \frac{1}{n} \sum_{i=1}^{n} \left( f(X_i) + (Y_i - f(X_i)) \frac{\xi_i}{\pi^{(\rho)}(X_i)} \right).$$

From the assumption, we have $\hat{\rho} = \rho^* + o_P(1)$. By the continuity of the budget-preserving path $\pi^{(\rho)}$, it follows that $\pi^{(\hat{\rho})}(X_i) = \pi^{(\rho^*)}(X_i) + o_P(1)$ for any $i \in \{1, \dots, n\}$. This, as a result, gives $\hat{\theta}^{\pi^{(\hat{\rho})}} = \hat{\theta}^{\pi^{(\rho^*)}} + o_P(1)$ by the continuity of $\hat{\theta}^{\pi^{(\rho)}}$.

It follows from Proposition 1 in [51] that we have

$$\sqrt{n} \left( \hat{\theta}^{\pi^{(\rho^*)}} - \theta^* \right) \xrightarrow{d} \mathcal{N}\left(0, \sigma_{\rho^*}^2\right), \quad \sigma_{\rho^*}^2 = \mathrm{Var}(\hat{\theta}^{\pi^{(\rho^*)}}).$$

Since $\hat{\theta}^{\pi^{(\hat{\rho})}} \xrightarrow{p} \hat{\theta}^{\pi^{(\rho^*)}}$,

$$\sqrt{n} \left( \hat{\theta}^{\pi^{(\hat{\rho})}} - \theta^* \right) \xrightarrow{d} \mathcal{N}\left(0, \sigma_{\rho^*}^2\right).$$

By the definition of $\rho^*$, $\rho^* = \arg\min_{\rho} \mathrm{Var}(\hat{\theta}^{\pi^{(\rho)}})$, we have

$$\sigma_{\rho^*}^2 = \mathrm{Var}(\hat{\theta}^{\pi^{(\rho^*)}}) \leq \min\{\mathrm{Var}(\hat{\theta}^{\pi^{(0)}}), \mathrm{Var}(\hat{\theta}^{\pi^{(1)}})\} = \min\{\sigma_0^2, \sigma_1^2\}.$$

This completes the proof.

## A.2  A sufficient condition for $\hat{\rho} = \rho^* + o_P(1)$

**Proposition 1.** Suppose $\hat{e}^2(X) = e^2(X) + o_P(1)$, and $\rho^*$ is unique. Suppose $\hat{e}(X)$ is uniformly upper bounded by $M > 0$. Suppose further that $\pi^{(\rho)}(X)$ is uniformly lower-bounded by $m > 0$, then we have $\hat{\rho} = \rho^* + o_P(1)$.

*Proof.* Denote

$$\mathcal{F} = \left\{ f_\rho(x) = \frac{\hat{e}^2(x)}{\pi^{(\rho)}(x)} : \rho \in [0, 1] \right\}.$$

We first show that $\mathcal{F}$ is a P-Glivenko-Cantelli class.

Since $\pi^{(\rho)}$ is continuous, and supported on $[0, 1]$, it is uniformly continuous on $[0, 1]$. Hence for any $\delta > 0$, there exists $\eta > 0$ such that $|\pi^{(\rho_1)}(X) - \pi^{(\rho_2)}(X)| \leq \frac{m^2}{M^2}\delta$ whenever $|\rho_1 - \rho_2| \leq \eta$. Now, we cover $[0, 1]$ with a grid $0 = \rho_0 < \rho_1 < \cdots < \rho_K = 1$, where $\rho_k - \rho_{k-1} = \eta$ for $k \leq K - 1$. Then, for any $\rho \in [\rho_{k-1}, \rho_k]$, we have

$$|f_\rho(x) - f_{\rho_{k-1}}(x)| = \left| \frac{\hat{e}^2(x)}{\pi^{(\rho)}(x)} - \frac{\hat{e}^2(x)}{\pi^{(\rho_{k-1})}(x)} \right| \leq \frac{M^2}{m^2} \left| \pi^{(\rho)}(x) - \pi^{(\rho_{k-1})}(x) \right| \leq \delta.$$

Hence $[f_{\rho_{k-1}} - \delta, f_{\rho_{k-1}} + \delta]$ is an $2\delta$-bracket in $L_1(P)$ that contains every $f_\rho$ with $\rho \in [\rho_{k-1}, \rho_k]$. So the bracketing number $N_{[]}$ is finite, $N_{[]}(2\delta, \mathcal{F}, L_1(P)) \leq K \leq \frac{1}{\eta} + 1 < \infty$. We thus conclude from the Blum-DeHardt theorem that $\mathcal{F}$ is a P-Glivenko-Cantelli class. Consequently, we have

$$\sup_{\rho \in [0,1]} \left| \frac{1}{n} \sum_{i=1}^{n} \frac{\hat{e}^2(X_i)}{\pi^{(\rho)}(X_i)} - \mathbb{E}\frac{\hat{e}^2(X)}{\pi^{(\rho)}(X)} \right| \xrightarrow{p} 0.$$

This implies that

$$\left| \inf_{\rho \in [0,1]} \frac{1}{n} \sum_{i=1}^{n} \frac{\hat{e}^2(X_i)}{\pi^{(\rho)}(X_i)} - \inf_{\rho \in [0,1]} \mathbb{E}\frac{\hat{e}^2(X)}{\pi^{(\rho)}(X)} \right| \xrightarrow{p} 0.$$

By definition, $\hat{\rho} = \arg\min_{\rho} \frac{1}{n}\sum_{i=1}^{n} \frac{\hat{e}^2(X_i)}{\pi^{(\rho)}(X_i)}$. Denote $S = \arg\min_{\rho} \mathbb{E}\frac{\hat{e}^2(X)}{\pi^{(\rho)}(X)}$. Then, by continuity of $\pi^{(\rho)}(X)$, we have $d(\hat{\rho}, S) \xrightarrow{P} 0$, for $d(\hat{\rho}, S) = \inf\{|\hat{\rho} - \hat{\rho}^*| : \hat{\rho}^* \in S\}$.

Now, for any $\hat{\rho}^* \in S$, we have

$$\mathbb{E}\frac{e^2(X)}{\pi^{(\hat{\rho}^*)}(X)} \le \mathbb{E}\frac{\hat{e}^2(X) + o_P(1)}{\pi^{(\hat{\rho}^*)}(X)}$$

$$\le \mathbb{E}\frac{\hat{e}^2(X)}{\pi^{(\rho^*)}(X)} + o_P(1)\mathbb{E}\frac{1}{\pi^{(\hat{\rho}^*)}(X)}$$

$$\le \mathbb{E}\frac{e^2(X) + o_P(1)}{\pi^{(\rho^*)}(X)} + o_P(1)\mathbb{E}\frac{1}{\pi^{(\hat{\rho}^*)}(X)}$$

$$= \mathbb{E}\frac{e^2(X)}{\pi^{(\rho^*)}(X)} + o_P(1)\mathbb{E}\left[\frac{1}{\pi^{(\rho^*)}(X)} + \frac{1}{\pi^{(\hat{\rho}^*)}(X)}\right]$$

$$= \mathbb{E}\frac{e^2(X)}{\pi^{(\rho^*)}(X)} + o_P(1).$$

Since

$$\mathrm{Var}(\hat{\theta}^{\pi^{(\rho)}}) = \mathbb{E}\left(\frac{e^2(X)}{\pi^{(\rho)}(X)}\right) + C,$$

where $C$ is a constant independent of $\rho$, we have

$$\mathrm{Var}(\hat{\theta}^{\pi^{(\hat{\rho}^*)}}) \le \mathrm{Var}(\hat{\theta}^{\pi^{(\rho^*)}}) + o_P(1).$$

On the other hand, by the definition of $\rho^*$,

$$\mathrm{Var}(\hat{\theta}^{\pi^{(\hat{\rho}^*)}}) \ge \mathrm{Var}(\hat{\theta}^{\pi^{(\rho^*)}})$$

also holds. Whence $\mathrm{Var}(\hat{\theta}^{\pi^{(\hat{\rho}^*)}}) \xrightarrow{P} \mathrm{Var}(\hat{\theta}^{\pi^{(\rho^*)}})$.

Since $\rho^*$ is the unique minimizer of $\mathrm{Var}(\hat{\theta}^{\pi^{(\rho)}})$, $\hat{\rho}^* \xrightarrow{P} \rho^*$ by continuity. Since $d(\hat{\rho}, S) \xrightarrow{P} 0$ and $\hat{\rho}^*$ is an arbitrary element in $S$, we immediately conclude that

$$\hat{\rho} \xrightarrow{P} \rho^*.$$

$\square$

### A.3 Proof of Theorem 2

By the definition of $\hat{\theta}^{\pi^{(\rho)}}$, we have

$$\hat{\theta}^{\pi^{(\rho)}} = \arg\min_{\theta} \frac{1}{n}\sum_{i=1}^{n}\left(\ell_{\theta,i}^f + \left(\ell_{\theta,i} - \ell_{\theta,i}^f\right)\frac{\xi_i}{\pi^{(\rho)}(X_i)}\right).$$

We assume $\hat{\rho} = \rho^* + o_P(1)$. By the continuity of the budget-preserving path $\pi^{(\rho)}$, it follows that $\pi^{(\hat{\rho})}(X_i) = \pi^{(\rho^*)}(X_i) + o_P(1)$ for any $i \in \{1, \ldots, n\}$. This, as a result, gives $\hat{\theta}^{\pi^{(\hat{\rho})}} = \hat{\theta}^{\pi^{(\rho^*)}} + o_P(1)$ by the continuity of $\ell_{\theta,i}^f + \left(\ell_{\theta,i} - \ell_{\theta,i}^f\right)\frac{\xi_i}{\pi^{(\rho)}(X_i)}$ with respect to $\theta$.

Given the assumption that $\hat{\theta}^{\pi^{(\rho^*)}} \xrightarrow{P} \theta^*$, from Theorem 1 in [51], we have

$$\sqrt{n}\left(\hat{\theta}^{\pi^{(\rho^*)}} - \theta^*\right) \xrightarrow{d} \mathcal{N}\left(0, \Sigma_{\rho^*}\right),$$

where $\Sigma_{\rho^*} = H_{\theta^*}^{-1}\mathrm{Var}\left(\nabla\ell_{\theta^*,i}^f + \left(\nabla\ell_{\theta^*,i} - \nabla\ell_{\theta^*,i}^f\right)\frac{\xi}{\pi^{(\rho^*)}(X_i)}\right)H_{\theta^*}^{-1}$.

Since $\hat{\theta}^{\pi^{(\hat{\rho})}} \xrightarrow{P} \hat{\theta}^{\pi^{(\rho^*)}}$,

$$\sqrt{n}\left(\hat{\theta}^{\pi^{(\hat{\rho})}} - \theta^*\right) \xrightarrow{d} \mathcal{N}\left(0, \Sigma_{\rho^*}\right).$$

The definition $\rho^* = \arg\min_{\rho} \Sigma_{jj}^{\pi^{(\rho)}}$ yields

$$\Sigma_{\rho^*,jj} = \Sigma_{jj}^{\pi^{(\rho^*)}} \leq \min\{\Sigma_{jj}^{\pi^{(0)}}, \Sigma_{jj}^{\pi^{(1)}}\} = \min\{\Sigma_{0,jj}, \Sigma_{1,jj}\}.$$

This completes the proof.

### A.4 Setting $\pi \propto \hat{e}$ leads to a trivial choice of $\hat{\rho} = 0$ when not incorporating robustness constraint

Starting from the variance estimate used in the optimization objective

$$\hat{\rho} = \arg\min_{\rho} \frac{1}{n} \sum_{i=1}^{n} \frac{\hat{e}^2(X_i)}{\pi^{(\rho)}(X_i)},$$

by the Cauchy-Schwarz inequality, for any $\pi$ such that $\sum_{i=1}^{n} \pi(X_i) = n_b$ (i.e. satisfying the budget constraint), $\sum_{i=1}^{n} \frac{\hat{e}^2(X_i)}{\pi(X_i)} \geq \frac{\left(\sum_{i=1}^{n} \hat{e}(X_i)\right)^2}{\sum_{i=1}^{n} \pi(X_i)} = \frac{\left(\sum_{i=1}^{n} \hat{e}(X_i)\right)^2}{n_b}$. The equality holds when $\pi \propto \hat{e}$, which corresponds to $\pi^{(\rho)}$ with $\rho = 0$.

## B   A natural family of budget-preserving paths

Among the diverse set of possible paths [28, 38], it is natural to consider *geodesic paths*, which are a family of "shortest paths."

**Definition 2** (Geodesic [7]). A curve $\gamma : I \rightarrow M$ from an interval $I \subseteq \mathbb{R}$ to a metric space $M$ with metric $d$ is a geodesic if there is a constant $v \geq 0$ such that for any $\rho \in I$ there is a neighborhood $J$ of $\rho$ in $I$ such that for any $\rho_1, \rho_2 \in J$ we have

$$d(\gamma(\rho_1), \gamma(\rho_2)) = v|\rho_1 - \rho_2|.$$

We revisit the examples from Section 2 and provide more geodesic paths.

In all the following examples, we assume $P$ and $Q$ have the same support.

**Example 3** (Linear path). The linear path, $\pi^{(\rho)} \propto (1-\rho)\pi + \rho\pi^{\text{unif}}$, is the geodesic path with respect to $d(P,Q) = \|P - Q\|$ with $v = \|\pi - \pi^{\text{unif}}\|$. Here, $\|\cdot\|$ is any norm.

**Example 4** (Geometric path). The geometric path, $\pi^{(\rho)} \propto \pi^{1-\rho}(\pi^{\text{unif}})^{\rho}$, is the geodesic path with respect to $d(P,Q) = \|\log P - \log Q\|$ with $v = \|\log \pi - \log \pi^{\text{unif}}\|$. Here, $\log$ is taken element-wise.

**Example 5** (Hellinger path). The Hellinger path, $\pi^{(\rho)} \propto \left((1-\rho)\sqrt{\pi} + \rho\sqrt{\pi^{\text{unif}}}\right)^2$, is the geodesic path with respect to $d(P,Q) = \|\sqrt{P} - \sqrt{Q}\|$ with $v = \|\sqrt{\pi} - \sqrt{\pi^{\text{unif}}}\|$. Here, the square root is taken element-wise.

**Note (more examples).** Some distance metrics may not have an analytical characterization for their corresponding geodesic path, such as the Wasserstein and Jensen-Shannon distances. However, it is computationally tractable to solve for a geodesic path numerically up to a tolerance margin for many well-defined distance metrics. For example, when computing the geodesic for the Jensen-Shannon distance, we can discretize the interval $[0,1]$ into $N$ segments so that $P_0 = P$ and $P_N = Q$, and we define a series of intermediate distributions $P_1, P_2, \ldots, P_{N-1}$. The task is then cast as an optimization problem: we minimize the total path length computed as the sum of the square roots of the Jensen-Shannon divergences between successive distributions, i.e., $\sum_{i=0}^{N-1} \sqrt{\text{JS}(P_i, P_{i+1})}$. Here, $\text{JS}(P\|Q) = \frac{1}{2}D(P\|M) + \frac{1}{2}D(Q\|M)$, where $M = \frac{1}{2}(P + Q)$. This is a constrained optimization problem and can be solved by standard gradient-based methods.

### B.1   Uniqueness of $\rho^*$

In Section 2, we saw that the uniqueness of the optimal $\rho^*$ and the consistency of $\hat{e}$ are sufficient conditions for the consistency of $\hat{\rho}$. In the case of all three budget-preseving paths from the previous section, it can be easily verified by computing the second derivative of $\text{Var}(\hat{\theta}^{\pi^{(\rho)}})$ that this variance is strictly convex and thus $\rho^*$ is unique. We include the corresponding proofs for completeness.

**Linear path.** We have $\pi^{(\rho)}(X) = (1 - \rho)\pi(X) + \rho\frac{n_b}{n}$. The problem of minimizing $\mathrm{Var}(\hat{\theta}^{\pi^{(\rho)}})$ is equivalent to

$$\arg\min_{\rho} \mathbb{E}\left[\frac{(Y - f(X))^2}{(1 - \rho)\pi(X) + \rho\frac{n_b}{n}}\right].$$

Denoting $g(\rho) = \mathbb{E}\left[\frac{(Y - f(X))^2}{(1 - \rho)\pi(X) + \rho\frac{n_b}{n}}\right]$, we have

$$g'(\rho) = \mathbb{E}\left[\frac{-(Y - f(X))^2\left(\frac{n_b}{n} - \pi(X)\right)}{\left((1 - \rho)\pi(X) + \rho\frac{n_b}{n}\right)^2}\right],$$

and

$$g''(\rho) = \mathbb{E}\left[\frac{2(Y - f(X))^2\left(\frac{n_b}{n} - \pi(X)\right)^2}{\left((1 - \rho)\pi(X) + \rho\frac{n_b}{n}\right)^3}\right].$$

Clearly, $g''(\rho) > 0$, which means that $g(\rho)$ is convex. Hence, there is a unique optimal value of $\rho$ in $[0, 1]$.

Notice that $g'(1) = \frac{n^2}{n_b^2}\mathbb{E}\left[(Y - f(X))^2\left(\pi(X) - \frac{n_b}{n}\right)\right]$. Hence, if $\mathbb{E}\left[(Y - f(X))^2\pi(X)\right] > \frac{n_b}{n}\mathbb{E}\left[(Y - f(X))^2\right]$, then $g'(1) > 0$, which implies that the optimal $\rho$ lies in $[0, 1)$.

**Geometric path.** Consider the path $\pi^{(\rho)}(X) \propto \pi(X)^{1-\rho}(\pi^{\mathrm{unif}})^\rho$; in particular, $\pi^{(\rho)}(X) = \frac{n_b}{n}\frac{\pi(X)^{1-\rho}}{\mathbb{E}[\pi(X)^{1-\rho}]}$.

Similar to the last example, we denote $g(\rho) = \mathbb{E}\left[\frac{(Y - f(X))^2}{\pi^{(\rho)}(X)}\right] = \frac{n}{n_b}\mathbb{E}\left[\frac{(Y - f(X))^2}{\pi(X)^{1-\rho}}\right]\mathbb{E}\left[\pi(X)^{1-\rho}\right]$. Then, we have

$$g'(\rho) = \frac{n}{n_b}\mathbb{E}\left[\frac{(Y - f(X))^2}{\pi(X)^{1-\rho}}\log\pi(X)\right]\mathbb{E}\left[\pi(X)^{1-\rho}\right] - \frac{n}{n_b}\mathbb{E}\left[\frac{(Y - f(X))^2}{\pi(X)^{1-\rho}}\right]\mathbb{E}\left[\pi(X)^{1-\rho}\log\pi(X)\right],$$

and

$$g''(\rho) = \frac{n}{n_b}\mathbb{E}\left[\frac{(Y - f(X))^2}{\pi(X)^{1-\rho}}\log^2\pi(X)\right]\mathbb{E}\left[\pi(X)^{1-\rho}\right] + \frac{n}{n_b}\mathbb{E}\left[\frac{(Y - f(X))^2}{\pi(X)^{1-\rho}}\right]\mathbb{E}\left[\pi(X)^{1-\rho}\log^2\pi(X)\right]$$
$$- 2\frac{n}{n_b}\mathbb{E}\left[\frac{(Y - f(X))^2}{\pi(X)^{1-\rho}}\log\pi(X)\right]\mathbb{E}\left[\pi(X)^{1-\rho}\log\pi(X)\right].$$

Since $(Y - f(X))^2 \geq 0$, $\pi(X) > 0$, and $\log^2\pi(X) \geq 0$, we have that

$$\mathbb{E}\left[\frac{(Y - f(X))^2}{\pi(X)^{1-\rho}}\log^2\pi(X)\right]\mathbb{E}\left[\pi(X)^{1-\rho}\right] + \mathbb{E}\left[\frac{(Y - f(X))^2}{\pi(X)^{1-\rho}}\right]\mathbb{E}\left[\pi(X)^{1-\rho}\log^2\pi(X)\right]$$

$$\geq 2\sqrt{\mathbb{E}\left[\frac{(Y - f(X))^2}{\pi(X)^{1-\rho}}\log^2\pi(X)\right]\mathbb{E}\left[\pi(X)^{1-\rho}\right]\mathbb{E}\left[\frac{(Y - f(X))^2}{\pi(X)^{1-\rho}}\right]\mathbb{E}\left[\pi(X)^{1-\rho}\log^2\pi(X)\right]}$$

$$= 2\sqrt{\mathbb{E}\left[\frac{(Y - f(X))^2}{\pi(X)^{1-\rho}}\log^2\pi(X)\right]\mathbb{E}\left[\frac{(Y - f(X))^2}{\pi(X)^{1-\rho}}\right]\mathbb{E}\left[\pi(X)^{1-\rho}\right]\mathbb{E}\left[\pi(X)^{1-\rho}\log^2\pi(X)\right]}$$

$$\geq 2\sqrt{\mathbb{E}^2\left[\frac{(Y - f(X))^2}{\pi(X)^{1-\rho}}\log\pi(X)\right]\mathbb{E}^2\left[\pi(X)^{1-\rho}\log\pi(X)\right]}$$

$$= \mathbb{E}\left[\frac{(Y - f(X))^2}{\pi(X)^{1-\rho}}\log\pi(X)\right]\mathbb{E}\left[\pi(X)^{1-\rho}\log\pi(X)\right].$$

The last inequality follows from the Cauchy-Schwarz inequality. Therefore, we have $g''(\rho) \geq 0$. Further, if $\pi(X) \neq \pi^{\text{unif}}$, the inequality is strict, which means $g(\rho)$ is convex. Thus, there is a unique optimal value of $\rho$ in $[0, 1]$.

**Hellinger path.** Suppose $P$ and $Q$ are two discrete distributions. The Hellinger distance between $P$ and $Q$ is $H(P, Q) = \frac{1}{\sqrt{2}} \|\sqrt{P} - \sqrt{Q}\|_2$. The geodesic connecting $\pi(X)$ and $\pi^{\text{unif}} = \frac{n_b}{n}$ is:

$$\pi^{(\rho)}(X) = \left( \frac{\sin((1-\rho)\beta)}{\sin \beta} \sqrt{\pi(X)} + \frac{\sin(\rho\beta)}{\sin \beta} \sqrt{\frac{n_b}{n}} \right)^2,$$

where $\beta = \arccos\left( \sum_{i=1}^{n} \sqrt{\frac{\pi(X_i)}{n} \cdot n_b} \right)$.

Similarly as above, minimizing the variance $\text{Var}(\hat{\theta}^{\pi^{(\rho)}})$ amounts to minimizing the function

$$g(\rho) = \mathbb{E}\left[ \frac{(Y - f(X))^2}{\left( \frac{\sin((1-\rho)\beta)}{\sin \beta} \sqrt{\pi(X)} + \frac{\sin(\rho\beta)}{\sin \beta} \sqrt{\frac{n_b}{n}} \right)^2} \right]$$

over $\rho$. The derivative $g'(\rho)$ is given by

$$-2\mathbb{E}\left[ (Y - f(X))^2 \left( \frac{\sin((1-\rho)\beta)}{\sin \beta} \sqrt{\pi(X)} + \frac{\sin(\rho\beta)}{\sin \beta} \sqrt{\frac{n_b}{n}} \right)^{-3} \left( -\beta \frac{\cos((1-\rho)\beta)}{\sin \beta} \sqrt{\pi(X)} + \beta \frac{\cos(\rho\beta)}{\sin \beta} \sqrt{\frac{n_b}{n}} \right) \right],$$

while the second $g''(\rho)$ is given by

$$\mathbb{E}\left[ (Y - f(X))^2 \left( \frac{\sin((1-\rho)\beta)}{\sin \beta} \sqrt{\pi(X)} + \frac{\sin(\rho\beta)}{\sin \beta} \sqrt{\frac{n_b}{n}} \right)^{-4} \left[ 6 \left( -\beta \frac{\cos((1-\rho)\beta)}{\sin \beta} \sqrt{\pi(X)} \right. \right. \right.$$

$$\left. \left. \left. + \beta \frac{\cos(\rho\beta)}{\sin \beta} \sqrt{\frac{n_b}{n}} \right)^2 + 2\beta^2 \left( \frac{\sin((1-\rho)\beta)}{\sin \beta} \sqrt{\pi(X)} + \frac{\sin(\rho\beta)}{\sin \beta} \sqrt{\frac{n_b}{n}} \right)^2 \right] \right] > 0.$$

Therefore, $g(\rho)$ is strictly convex, and there is a unique optimal value of $\rho$ in $[0, 1]$.

## C   Perturbed model errors after robust optimization

It is natural to choose the constraint $\mathcal{C}$ by upper-bounding the norm of $\epsilon$. Our default choice is the $\ell_2$ norm, i.e. $\|\epsilon\|_2 \leq c$. The $\ell_2$ norm can be roughly thought of as controlling the variance of the errors in $\hat{e}^2$. In particular, imagine $\hat{e}^2(X_i)$ can be viewed as a noisy version of $e^2(X_i)$: $\hat{e}^2(X_i) = e^2(X_i) + \xi_i$, where the $(X_i, \xi_i)$ pairs are i.i.d. and $\xi_i$ have mean zero. Then, by concentration, $\|\epsilon\|_2^2 \approx \sum_i \text{Var}(\xi_i)$.

In Figure 8 we illustrate how robust optimization over the $\ell_2$ set $\mathcal{C}$ recovers errors $\hat{e}^2(X_i) + \epsilon_i$ that are much closer to $e^2(X_i)$ than simply using $\hat{e}^2(X_i)$.

## D   A toy example: choice of $\mathcal{C}$

A simple $\ell_2$ norm constraint may not always be the most powerful choice of $C$. Zooming out, our method can in principle be combined with *any* choice of $C$, including one where we learn regions of the space where scores are systematically overconfident or underconfident. At a high level, our method (1) learns $C$ (in our experiment, the "learning" is a simple fitting of $c$ through cross-validation), and (2) solves a robust optimization problem with $C$ in place. Your suggestion is an interesting choice of step (1).

We developed a dataset featuring a central "hard" region ($|X| \leq 2$) flanked by two "easy" regions ($2 < |X| < 5$). In the easy regions, error data was sampled from $\mathcal{N}(2, 0.05)$. In the hard region, error was drawn from $\mathcal{N}(1, 0.25)$. The estimator of error, $\hat{e}(X)$, is designed to underestimate the

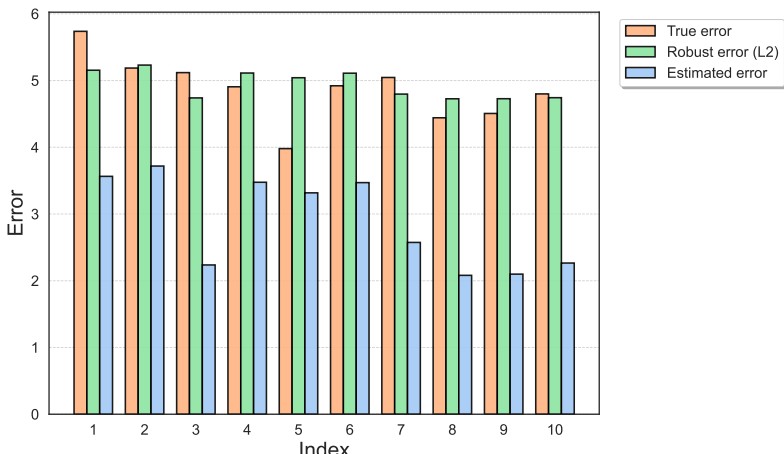

Figure 8: **Perturbed errors** $\hat{e}^2(X_i) + \epsilon_i$ **vs naive errors** $\hat{e}^2(X_i)$ **with** $\ell_2$ **constraint** $\mathcal{C}$. We consider a regime where we underestimate the true error (for example, due to the model being overconfident). We let $e(X_i) \sim \mathcal{N}(5, 0.25)$ and $\hat{e}(X_i) \sim \mathcal{N}(3, 0.25)$, and $\pi^{(\rho)}$ is the linear path with $\rho = 0.5$. The robustness constraint is $\mathcal{C} = \{\epsilon : \|\epsilon\|_2 \leq 50\}$. Each index $i$ corresponds to one sample $X_i$. The robust error (green bar) is the error after perturbation, $\hat{e}^2(X_i) + \epsilon_i$, and the estimated error (blue bar) is the error before perturbation, $\hat{e}^2(X_i)$. The robust errors are much closer to the estimated errors.

error in the hard region and overestimate the error in the easy region. Specifically, $\hat{\epsilon}(X) = 0.5$ for $|X| \leq 2$, and $\hat{\epsilon}(X) = 2.5$ otherwise.

Subsequently, we trained a meta-classifier, a gradient boost classifier, $h(X)$, to identify these regions solely based on the performance of $\hat{\epsilon}(X)$, without prior knowledge of the region boundaries.

This approach proved highly effective, with the meta-classifier achieving over 99% accuracy in identifying the regions. This demonstrates our success in learning the error regions and enables us to separate the constraint set $C$ based on these distinctions. For instance, $C$ can be defined as $\|\epsilon_{\text{easy}}\|_2 \leq c_{\text{easy}}$ for the easy region ($2 < |X| < 5$) and $\|\epsilon_{\text{hard}}\|_2 \leq c_{\text{hard}}$ for the hard region ($|X| \leq 2$). Or even simpler, we can only optimize over hard regions, i.e. $c_{\text{easy}} = 0$. While these regions' dimensions are not fixed and depend on $X$, this presents no practical difficulties because we have complete information about $X$.

Next, we compared this structured constraint with the global constraint. Here, for the structured constraint, we only optimize over the hard region. The following table shows the result when $n = 7000$, $n_h = 1400$, and $\pi \propto \hat{\epsilon}$.

| Method | ESS | ESS Gain (%) |
|---|---|---|
| Uniform | 1400 | 0.00% |
| Active | 1213 | -13.3% |
| Robust active (global) | 1491 | 6.5% |
| Robust active (structured) | 1495 | 6.8% |

We found that incorporating the structured constraint provided a slight gain in ESS over the global constraint while reducing the constraint size ($c_{\text{global}} = 85$ vs. $c_{\text{hard}} = 75$). This suggests that a more focused perturbation can be beneficial when we have strong knowledge of confident regions. However, we note that the global constraint remains a simple and practical approach given the limited gain.

# E   Additional experimental results

## E.1   Plots with coverage

In this subsection, we provide figures corresponding to the figures in the main text, where in addition to the effective sample size we also plot coverage.

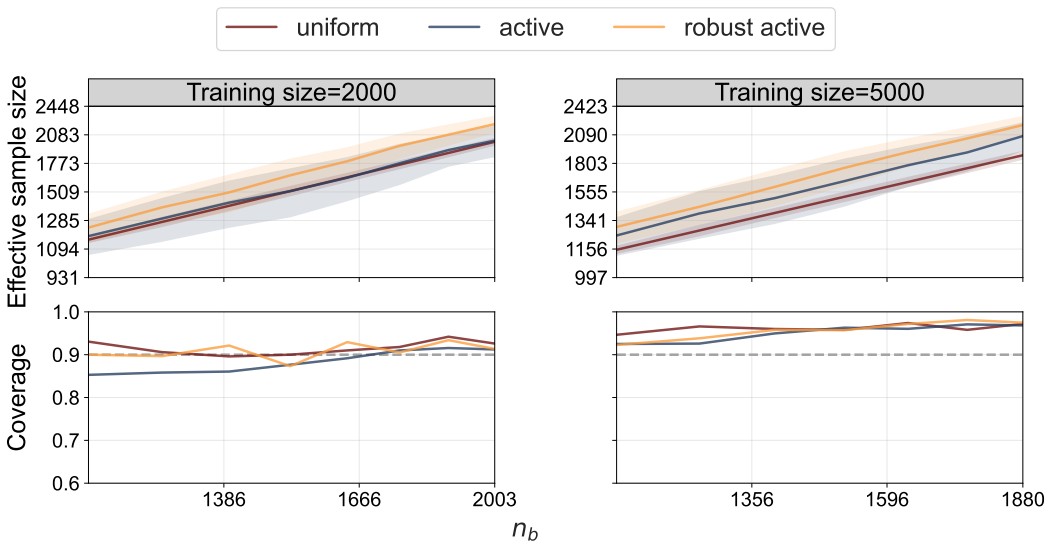

Figure 9: **Effective sample size and coverage on Pew post-election survey data**, for different dataset sizes used to train $f$. We compare uniform, active, and robust active sampling, for different values of the sampling budget $n_b$. The target of inference is the approval rate of a presidential candidate.

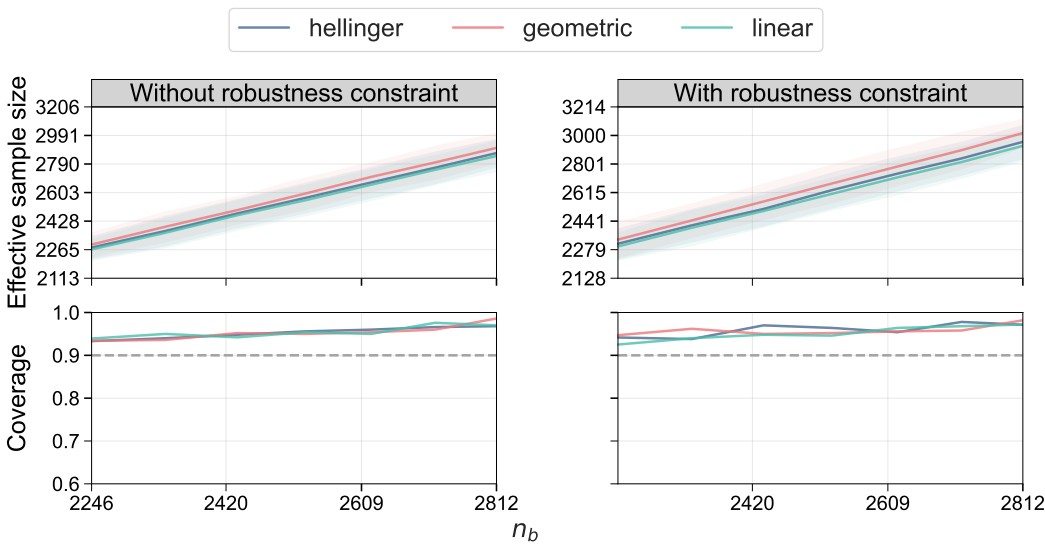

Figure 10: **Effective sample size and coverage for different budget-preserving paths on Pew post-election survey data**, without (left) and with (right) a robustness constraint $\mathcal{C}$. In both cases, the geometric path leads to the largest effective sample size. The target of inference is the same as in Figure 3.

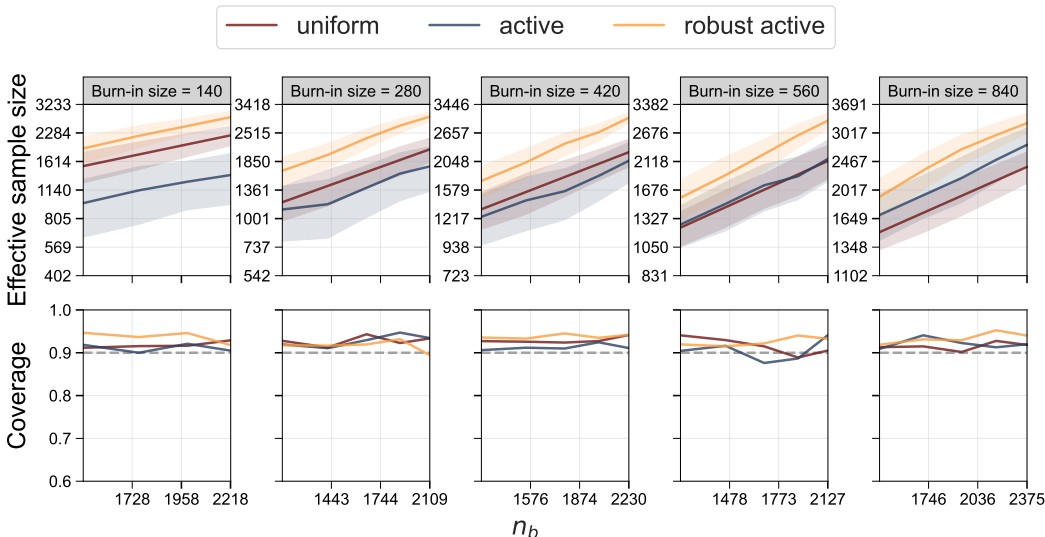

Figure 11: **Effective sample size and coverage on US Census data**, for varying burn-in dataset sizes. We compare uniform, active, and robust active sampling, for different values of the sampling budget $n_b$. The target of inference is the relationship between age and income, estimated via a linear regression.

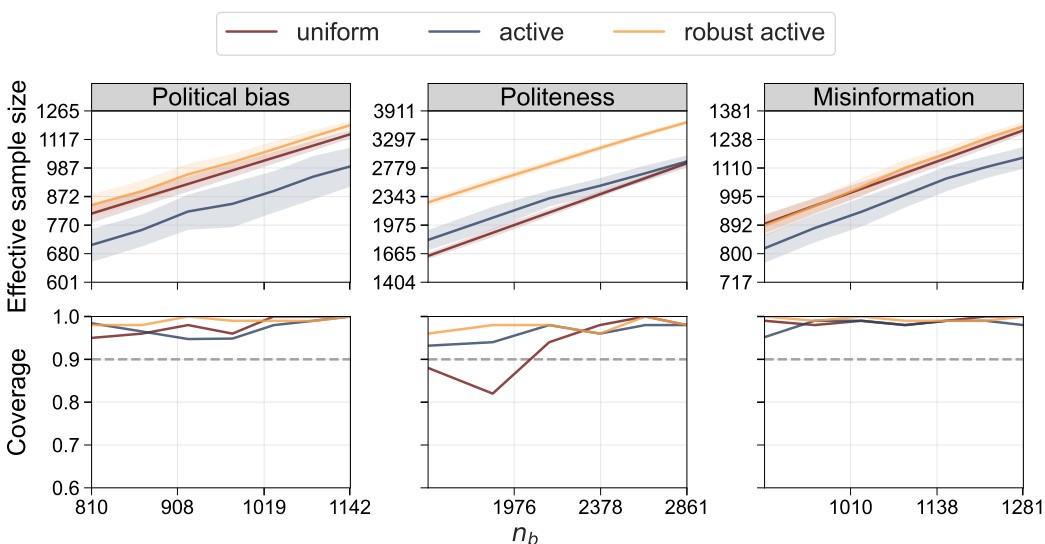

Figure 12: **Effective sample size and coverage on social science text annotation datasets**. We compare uniform, active, and robust active sampling, for different values of the sampling budget $n_b$. The targets of inference are (left to right) the prevalence of right-leaning political bias, the relationship between hedging and politeness, and the prevalence of misinformation.

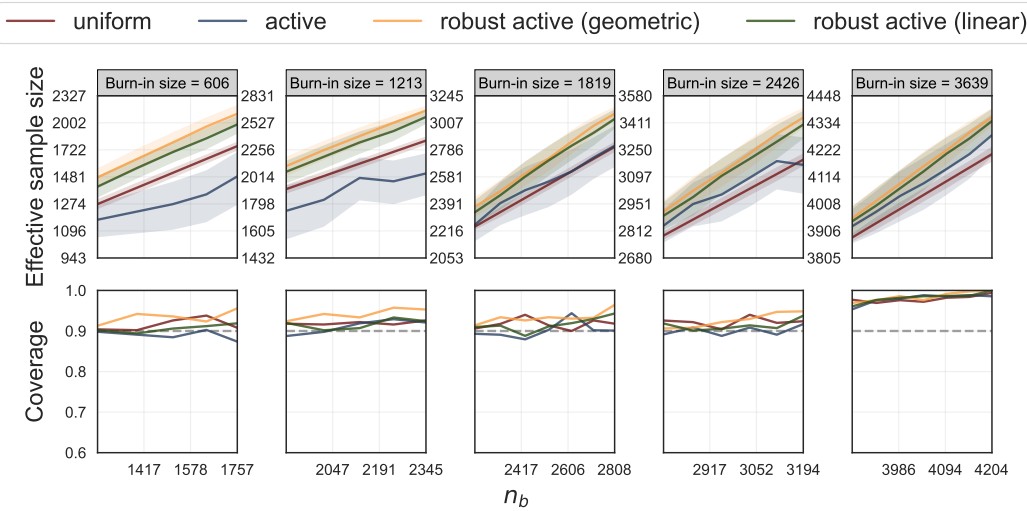

Figure 13: **Effective sample size (top) and coverage (bottom) on Pew post-election survey data**, for varying burn-in dataset sizes with respect to different proportions of the data. We compare uniform, active, and robust active sampling with geometric and linear paths, for different values of the sampling budget $n_b$. The target of inference is the approval rate of a presidential candidate.

## E.2 Burn-in size v.s. robustness constraint $\mathcal{C}$

In addition to optimized $\rho_{\text{robust}}$ along the path, we also provided the optimized value $c$ in the robustness constraint $\mathcal{C} = \{\boldsymbol{\epsilon} : \|\boldsymbol{\epsilon}\|_2 \leq c\}$. As expected, we observe a more conservative constraint when the errors are poorly estimated.

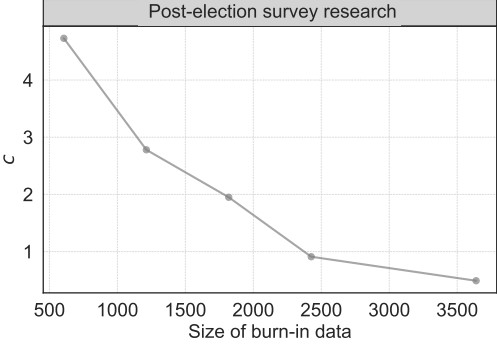

Figure 14: **Optimized value $c$ along the geometric path** as a function of the size of the burn-in data for the post-election survey data.

## E.3 Sensitivity to step size

When we solve the optimization problem 4, we employ a grid search for $\rho$ in the outer loop. We conducted experiments to explore different step sizes of the grid search and confirmed the robustness of our results to step-size selection, as shown below. The ESS gap between these two estimators is minimal, and both significantly outperform uniform and active baselines.

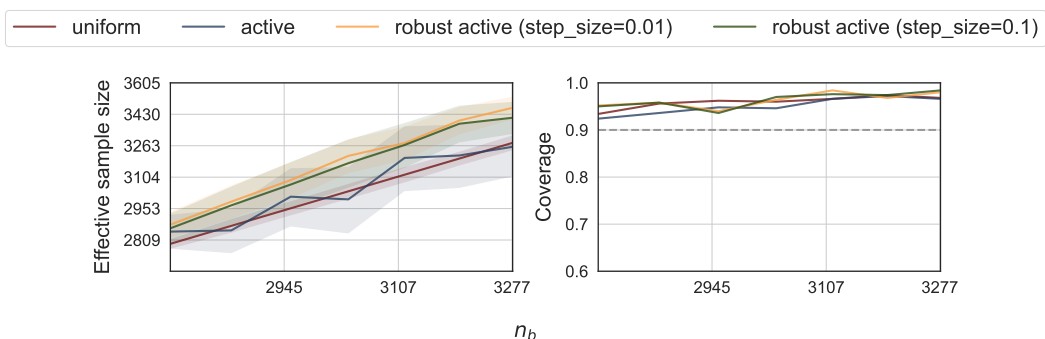

Figure 15: **Effective sample size on Pew post-election survey data**, for different step sizes in grid search for $\rho$. We compare uniform, active, and robust active sampling with grid search step sizes of 0.01 and 0.1. The target of inference is the approcal rate of a presidential candidate.

