# OpenReview forum: "Robust Sampling for Active Statistical Inference"
_NeurIPS.cc/2025/Conference — NeurIPS 2025 poster_

### Official Review · Reviewer_usPt · 2025-06-20

**Clarity:** 4
**Significance:** 3
**Originality:** 4
**Rating:** 5
**Confidence:** 3

**Summary:**

The authors’ goal is to increase the efficiency of active sampling policies used in the context of active statistical inference: Inaccurate predictive models can give poor uncertainty estimates, which can result in supposedly efficient sampling methods (e.g., uncertainty sampling) having a larger sample efficiency than uniform sampling. To address this, the authors propose a method which smoothly interpolates between an active and uniform sampling policy, where the “optimal” interpolation is identified in a data-driven way.

**Questions:**

**Sensitivity to choice of budget-preserving path:** In Figure 4, the linear and Hellinger budget-preserving paths appear to perform about as well as uniform sampling. In general, how sensitive is the method’s performance to the choice of budget-preserving path? In practice, how would a user select a good budget-preserving path?

**Uncertainty set:**
Theorem 2 requires that $\hat{\rho} = \rho^\star + o_P(1)$. Does the degree of misspecification of $\hat{e}$ affect whether this condition is violated or not? How does incorporation of the uncertainty set affect whether/when this condition is violated?

In section 2.2, it is mentioned that a permissive uncertainty set results in a large value of $\hat{\rho}$, which is intuitive: A misspecified predictive model is less likely to produce a reliable active sampling policy. Are there advantages to solving the robust optimization problem instead of, for instance, adding a penalty for small values of $\rho$ to the original optimization problem (line 131)?

**Ethical Concerns:**

["NO or VERY MINOR ethics concerns only"]

**Final Justification:**

The submission tackles a somewhat significant problem and appears original and technically sound. My main concern was the effectiveness of the proposed method. The authors' rebuttal included additional discussion and experimental results, which (i) clarified the implications of the theoretical results and range of the method’s behavior, and (ii) provided concrete guidance about how to select the budget-preserving path. I believe including this discussion and these results in the paper will enhance the paper’s contributions.

**Limitations:**

Yes. While the authors mention the sensitivity of their method to the choice of budget-preserving path, more details on the extent of this sensitivity could be provided.

**Quality:**

3

**Strengths And Weaknesses:**

The submission tackles a somewhat significant problem: Predictive (surrogate) models are used to complement scarce data in many practical applications, and robustness of policies informed by such models to the potentially unreliable uncertainty estimates they produce is important in such applications. The scope of the paper is limited to the design of robust active sampling policies; active sampling is useful especially in the types of data-poor settings in which predictive-powered inference is applied.

The submission appears original, technically sound, and the claims well-supported by both theoretical analysis and experimental results. It is exceptionally clearly written and well organized.

My main concern regards the effectiveness of the authors’ method: I think the paper could benefit from more discussion of the method’s sensitivity to user-specified aspects, in particular the choice of budget-preserving path.

---

> ### Author Rebuttal · Authors · 2025-07-31
>
> We really appreciate your comprehensive feedback. We have responded to each point individually below, and hope our responses adequately resolve your concerns.
>
> ---
>
> > “Sensitivity to choice of budget-preserving path: In Figure 4, the linear and Hellinger budget-preserving paths appear to perform about as well as uniform sampling. In general, how sensitive is the method’s performance to the choice of budget-preserving path? In practice, how would a user select a good budget-preserving path?”
>
>
> Our central claim is that our procedure yields an estimator that performs no worse than either the uniform or active baseline, and that the gains can in many cases be significant. We cannot guarantee the extent of performance gain for any specific problem.
>
> However, to further evaluate the possible path choices, we added a new experiment comparing the geometric and linear paths against the active and uniform baselines in the setting of Figure 3. The findings presented below support our central claim above: the linear path consistently performs no worse than either of the two baselines, even if it does not achieve the same performance as the geometric path.
>
> Burn-in size 606:
>
> | $n_b$ | Robust active (geometric) | Robust active (linear) | Active | Uniform |
> |-------|---------------------------|------------------------|--------|--------|
> |  1396    |            1511               |             1434         |     1195   |   1396    |
> | 1518    |                 1652          |              1569         |     1264   |     1518   |
> | 1760    |                  1932         |              1836         |     1422   |     1760   |
> | 2003   |                    2205       |               2097         |    1652   |     2003   |
>
> Burn-in size 1213:
>
> | $n_b$ | Robust active (geometric) | Robust active (linear) | Active | Uniform |
> |-------|---------------------------|------------------------|--------|--------|
> |  1807    |            1896               |          1844            |     1744   |   1807    |
> | 2023    |                 2142         |               2073        |     1954   |     2023   |
> | 2239    |                  2393         |               2286        |     2058   |     2239   |
> | 2455   |                    2641       |             2519           |    2218   |     2455   |
>
> Burn-in size 1819:
>
> | $n_b$ | Robust active (geometric) | Robust active (linear) | Active | Uniform |
> |-------|---------------------------|------------------------|--------|--------|
> |  2340    |            2418               |            2358           |     2288   |   2340    |
> | 2529    |                 2620          |               2553         |     2481   |     2529   |
> | 2718    |                  2854         |                2765        |     2646   |     2718   |
> | 2907   |                   3066        |             2963           |     2823   |     2907   |
>
> Burn-in size 2426:
>
> | $n_b$ | Robust active (geometric) | Robust active (linear) | Active | Uniform |
> |-------|---------------------------|------------------------|--------|--------|
> |  2793    |            2866               |         2840          |     2796   |   2793    |
> | 2955   |               3064          |                3033       |     2982   |     2955   |
> | 3117    |               3265         |              3189        |     3131   |     3117   |
> | 3279   |                3458        |             3364        |     3287  |     3279   |
>
> Burn-in size 3639:
>
> | $n_b$ | Robust active (geometric) | Robust active (linear) | Active | Uniform |
> |-------|---------------------------|------------------------|--------|--------|
> |  3881    |            3963               |           3938        |     3923   |   3881    |
> | 3935   |               4024         |              4022         |     4010   |     3935   |
> | 4043    |               4166         |               4147       |     4126   |     4043   |
> | 4151   |                4308        |               4285      |     4251  |     4151  |
>
>
>
> ---
>
> > “Uncertainty set: Theorem 2 requires that $\hat{\rho} = \rho^*+o_p(1)$. Does the degree of misspecification of $\hat{e}$ affect whether this condition is violated or not? How does incorporation of the uncertainty set affect whether/when this condition is violated?”
>
>
>
> Good question: the consistency of $\hat{\rho}$ is indeed affected by the degree of misspecification in $\hat{e}$. When $\hat{e}$ is consistent for $e$, the consistency of $\hat{\rho}$ can be shown under mild conditions (as detailed in the Appendix). However, a high degree of misspecification in $\hat{e}$ leads to $\hat{\rho}$ being inconsistent, which necessitates the incorporation of a robustness constraint.
>
> The size of the constraint set influences $\rho_{\mathrm{robust}}$: a small set results in $\rho_{\mathrm{robust}}$ being close to 0 (active), while a large set yields $\rho_{\mathrm{robust}}$ close to 1 (uniform). If the constraint set is too large, we may lose the guarantee of convergence to $\rho^*$, but by appropriately choosing the constraint set $\mathcal{C}$ we still offer improved performance over the uniform baseline, as demonstrated empirically.
>
>
> ---
>
> > "In section 2.2, it is mentioned that a permissive uncertainty set results in a large value of $\hat{\rho}$, which is intuitive: A misspecified predictive model is less likely to produce a reliable active sampling policy. Are there advantages to solving the robust optimization problem instead of, for instance, adding a penalty for small values of \rho to the original optimization problem (line 131)?"
>
>
> This is a very interesting suggestion. Both approaches are interpretable and computationally efficient. We feel that our approach is more interpretable, since it directly incorporates a robustness wrapper around the source of the misspecification, which are the error estimates. The regularization approach does this more implicitly. Tuning the regularization parameter also feels potentially more involved than determining the robustness constraint $c$, which is simply based on measuring the error of $\hat e$ on a small dataset. We will add a brief discussion of this alternative in the revised version of the manuscript. Thank you for suggesting it!
>
> ---

---

> > ### Comment · Reviewer_usPt · 2025-08-03
> >
> > Thank you for the clarifying response. The authors' rebuttal highlights the paper's theoretical and empirical results demonstrating that the proposed estimator never performs worse, and often performs better, than the uniform and active baselines. The development of such an estimator is a significant contribution. However, I think the contribution of the paper would be enhanced by more discussion of the conditions under which the proposed estimator outperforms the baselines. For example, it seems that in your experiments the geometric path always leads to better performance than other choices of budget-preserving path. Do you have any thoughts as to what may account for the empirical advantage of the geometric path, and how consistent this advantage is? Are there settings in which a linear or Hellinger path might lead to better performance? Given this evidence, could you offer brief advice to users about how to select a budget-preserving path (or a small set of paths to then evaluate on a held-out dataset, as suggested in the text)?

---

> > > ### Author Response · Authors · 2025-08-05
> > > **Further Response to Reviewer usPt**
> > >
> > > Dear Reviewer,
> > >
> > > Thank you for engaging in the discussion. We agree that other paths may be superior in specific settings. Developing a deeper theoretical understanding of which problem properties favor certain paths is an exciting direction for future work.
> > >
> > > Based on our existing findings, this is our proposal:
> > > 1. As a practical default, we recommend using the geometric path. It has been stress-tested and has consistently demonstrated strong performance in our evaluations. We believe this is a good tradeoff between simplicity and performance.
> > > 2. For improved performance, use a data-driven approach. As suggested, a practitioner can evaluate a small set of candidate paths (e.g., geometric, linear, Hellinger) on a held-out dataset and select the path that minimizes the estimated variance (see line 250).
> > >
> > > We will incorporate this guidance into the revised manuscript. Thank you again for the comments!

---

> > > > ### Comment · Reviewer_usPt · 2025-08-08
> > > >
> > > > Thank you for the discussion of the selection of budget-preserving path. In conjunction with the experimental results in the paper, the additional experimental results above nicely illustrate the implications of the theoretical results and range of the method’s behavior (showing both a case where the method consistently outperforms the baselines and a case where it performs no worse but not substantially better than the baselines), and highlights the method’s sensitivity to the choice of budget-preserving path. I think the following two additions to the manuscript would help readers better assess when and how to apply the proposed method: (i) discussion of the additional experimental results, and (ii) a brief subsection or paragraph with the above discussion of selection of budget-preserving path. I maintain my recommendation to accept the paper.

---

### Official Review · Reviewer_myd5 · 2025-06-25

**Clarity:** 3
**Significance:** 2
**Originality:** 2
**Rating:** 3
**Confidence:** 4

**Summary:**

This paper builds upon the recently introduced area of active statistical inference, in which the goal is to formulate a sampling policy for the collection of a labeled dataset that will be used in conjunction with a predictive model to estimate some statistical quantity of interest. Specifically, the paper formulates a robust active statistical inference problem where the goal is to find a sampling policy that is no worse than two given reference policies in terms of variance. The paper focuses on the case where one of the policies is the naive uniform sampling policy.


The paper's solution to this problem is to consider a path of policies that interpolate between the given policies and to choose the policy along the path that minimizes the estimate of the variance. To account for potential model misspecification, the paper considers a robust version of their estimator that allows for a perturbation of the variance estimator.


The paper goes on to apply their techniques to real-world data, showing that their method outperforms both uniform and non-robust active sampling strategies.

**Questions:**

- Empirically, how important is the robustness criterion for the proposed method?

- What do the results of the paper look like across multiple random seeds?

- Do you have any intuition for why one interpolation method should be preferred over another?

**Ethical Concerns:**

["NO or VERY MINOR ethics concerns only"]

**Final Justification:**

After reading the other reviews and going through the author rebuttals, I still believe this is a borderline paper.

The novelty of the paper beyond the original active statistical inference paper is somewhat limited, and the paper in its current form does not do a great job of delineating what is truly novel and what was known before. For the actual novelty of the work, the authors introduce two ideas: alternate types of interpolating paths and a robust version of the variance estimator. On both of these ideas, the paper in its current form feels like it has only performed a surface-level investigation.

For the alternate types of interpolating paths, the paper presents 3 out of a literally infinite class and then uses a small number of simulations to justify settling on one. There is no attempt to answer the question of why that particular scheme works better than the others.

For the robust version of the variance estimator, it seems like a few versions were tested out in the writing of the paper, but ultimately just one type is presented with no real justification. Why does that form of robustness seem to work so well for this problem? What types of misspecifications are being fixed here? As my model class gets closer to the ground truth, what is the effect of adding/removing this robustness? Is it a bias issue with the model or a variance issue with its predictions?

I feel that diving into these kinds of fundamental questions would greatly improve the paper.

**Limitations:**

Yes.

**Quality:**

2

**Strengths And Weaknesses:**

Strengths:

- This paper considers an interesting problem at the intersection of robustness and active data collection. Although some aspects of the problem were already covered in the original active statistical inference paper (see below), the paper considers a broader family of interpolation techniques along with a robustness modification to the optimization problem.


- The paper evaluates the proposed methodology on 5 different datasets, covering a variety of inference tasks and prediction methods. Overall, their methods enjoy better empirical performance than the baseline approaches.

Weaknesses:

- The novel contributions of the paper need to more clearly delineated. As the paper is written, one might think that this paper is the first to consider the concept of finding the minimum variance interpolating sampling policy. However, this idea was already proposed in the original Zrnic and Candes active statistical inference paper, albeit only for the case of linear interpolants and without any robustness constraints. The contributions of this paper seem to be that it (a) expands the set of interpolating paths, (b) that it adds a robustness constraint to the variance estimator, and (c) that it gives a more thorough evaluation of uniform v.s. active v.s. robust active.


- It is hard to tell how useful the robustness constraint is. There is one figure comparing the performance with and without the robustness constraint, confusingly shown twice (Figure 4 and Figure 10). However, because the x and y-axes are not the same between the two, I can't really eyeball how useful the robustness constraint is. There should be several figures comparing with and without robustness for the various path interpolants on ALL the datasets, and they should also be on the SAME axes (maybe use dotted and solid lines). It would also be helpful to see what happens as the robustness hyperparameter c is varied.


- There is no mention of multiple runs/random seeds in the paper nor are there any confidence intervals/error bars in the plots. Given how much randomness is involved in this paper (initial training set, random label selection according to the policy, etc), it would be good to see how confident we should be in the results of the paper.


- As the paper currently stands, there is no guidance on how to choose the interpolating path. There is one experiment showing that the geometric path empirically outperforms the other two proposed paths on the Pew survey dataset, but there's no reasoning given. Moreover, it's worth pointing out that one could actually consider many types of paths simultaneously and choose the best path/interpolating fraction based on minimizing the variance estimator.

---

> ### Author Rebuttal · Authors · 2025-07-31
>
> Thank you for your detailed review and helpful feedback. We have addressed each of your points below. Please let us know if these responses adequately address your concerns.
>
> ---
>
> > “The novel contributions of the paper need to more clearly delineated. As the paper is written, one might think that this paper is the first to consider the concept of finding the minimum variance interpolating sampling policy. However, this idea was already proposed in the original Zrnic and Candes active statistical inference paper, albeit only for the case of linear interpolants and without any robustness constraints. The contributions of this paper seem to be that it (a) expands the set of interpolating paths, (b) that it adds a robustness constraint to the variance estimator, and (c) that it gives a more thorough evaluation of uniform v.s. active v.s. robust active.”
>
>
>
> Thank you for this observation. We acknowledge that we missed including a discussion of the robustness proposal from [1]. We will make sure to clarify the distinction in the revised version of the paper. The proposal from [1] assumes access to historical data to tune the linear interpolation coefficient. Otherwise, it selects a default value for the coefficient, such as 0.5, which does not have the guarantee of outperforming uniform and active sampling. Our analysis is far more thorough and systematic, expanding the set of interpolating paths, not requiring historical data but incorporating a burn-in period, and adding in a robustness constraint. These are all crucial for the practicality and reliability of the method. For example, the robustness constraint is critical: for natural choices of $\pi$ proposed in [1], tuning the interpolation coefficient *only* would result in trivial sampling rules (see paragraph starting in line 230).
>
> ---
>
> > “It is hard to tell how useful the robustness constraint is. There is one figure comparing the performance with and without the robustness constraint, confusingly shown twice (Figure 4 and Figure 10). However, because the x and y-axes are not the same between the two, I can't really eyeball how useful the robustness constraint is. There should be several figures comparing with and without robustness for the various path interpolants on ALL the datasets, and they should also be on the SAME axes (maybe use dotted and solid lines). It would also be helpful to see what happens as the robustness hyperparameter c is varied.”
>
>
>
> This is a very good suggestion. (And thank you for pointing out the repeated figure; Figure 10 includes the coverage plots accompanying Figure 4 due to the space limit of the main paper.) The robustness constraint is indeed crucial. As discussed in Section 4.1 (line 230), if we set $\pi \propto \hat{e}$, without the additional robust optimization step, the method would trivially estimate $\hat{\rho} = 0$. This means that in all experiments that rely on this choice of $\pi$, *the plain active baseline is equivalent to not including the robustness constraint*. To give some numbers, here are some results from Figure 5:
>
> | $n_b$ | With robustness constraint | Without robustness constraint | Active | Uniform  |
> |-------|---------------------------|------------------------|--------|--------|
> |  1498    |            1921               |            1691          |     1691   |   1498    |
> | 1717    |                 2317          |               1944         |     1944   |     1717   |
> | 1936    |                 2689         |                2195        |     2195  |     1936   |
> | 2156   |                   2971       |             2495          |     2495   |     2156   |
> | 2375   |            3233               |             2775           |     2775   |     2375   |
>
> Below we give an additional example when $\pi$ is not proportional to $\hat{e}$, specifically in the setting of Figure 4, where the burn-in size is 1819, and $\pi \propto \min\\{f(X), 1-f(X)\\}$:
>
> | $n_b$ | With robustness constraint | Without robustness constraint | Active | Uniform |
> |-------|---------------------------|------------------------|--------|--------|
> |  2339    |            2464              |             2406         |     2337   |   2339  |
> | 2433    |              2558         |              2491         |     2439   |     2433   |
> | 2622    |              2753           |              2696         |     2605   |     2622   |
> | 2811   |                 2974       |               2869         |    2803   |     2811   |
>
> We will include the above results in the revision. Thank you again for this valuable suggestion.
>
> ---
>
>
> > “There is no mention of multiple runs/random seeds in the paper nor are there any confidence intervals/error bars in the plots. Given how much randomness is involved in this paper (initial training set, random label selection according to the policy, etc), it would be good to see how confident we should be in the results of the paper.”
>
>
>
> We appreciate and agree with this suggestion. We average all metrics over 500 trials. For the coverage plots, given that they are obtained by averaging 500 binary indicators, the standard deviation does not add any information, since it equals $\sqrt{\mathrm{coverage}(1-\mathrm{coverage})}$, and should not be plotted (as per standard convention). However, we will plot the standard deviation of the effective sample size everywhere. Below we include the numbers for the problem of collecting politeness annotations with LLMs.
>
> | $n_b$  | Robust active ESS | Robust active Std | Active ESS | Active Std | Uniform ESS | Uniform Std |
> |------|-------------------|-------------------|------------|------------|-------------|-------------|
> | 1643 | 2263              | 51.9              | 1808       | 109.0      | 1643        | 30.3        |
> | 1887 | 2563              | 63.2              | 2065       | 139.4      | 1887        | 47.7        |
> | 2130 | 2849              | 56.8              | 2319       | 111.7      | 2130        | 48.1        |
> | 2374 | 3132              | 55.6              | 2499       | 118.6      | 2374        | 40.2        |
> | 2617 | 3402              | 49.4              | 2693       | 128.5      | 2617        | 37.3        |
> | 2861 | 3654              | 47.9              | 2891       | 105.8      | 2861        | 39.6        |
>
>
>
> ---
>
> > “As the paper currently stands, there is no guidance on how to choose the interpolating path. There is one experiment showing that the geometric path empirically outperforms the other two proposed paths on the Pew survey dataset, but there's no reasoning given. Moreover, it's worth pointing out that one could actually consider many types of paths simultaneously and choose the best path/interpolating fraction based on minimizing the variance estimator.”
>
>
>
> We have stress tested the geometric path on multiple datasets (with different modalities, uncertainty scores, etc) and found it to be the best choice among the considered paths. While we cannot guarantee that this will be the case everywhere, our main goal is to provide a method that consistently performs no worse than uniform or plain active sampling. Across *all* datasets, interpolating paths, uncertainty measures, we have found this core message to be true. Developing a deeper understanding around what properties of the problem determine the right interpolating path is a non-trivial and exciting future direction. We agree that it would be beneficial to simultaneously consider multiple paths and select the best one automatically. The risk is that this might be somewhat less practical, but it would be worthwhile to study the performance gains and see if they justify the extra effort. For now, given our current findings, we recommend the geometric path as a simple, practical default.
>
> ---
>
>
> [1] Zrnic, Tijana, and Emmanuel Candes. "Active Statistical Inference." International Conference on Machine Learning. PMLR, 2024.

---

> > ### Comment · Reviewer_myd5 · 2025-08-04
> >
> > Thank you to the authors for the clarifications. A few points:
> >
> > Regarding the statement that setting $\pi = \hat{e}/C$ results in a trivial choice of $\rho$,
> > 1. Do you have a formal proof of this? I can believe that this does occur, but I don't immediately see it.
> > 2. It's a little confusing to see a justification for the robustness constraint arising from trying to avoid the trivial setting of $\rho$. The fact that this particular choice of $\pi$ results in $\rho=0$ is more of a reflection of the fact that this choice of $\pi$ is supposed to minimize a particular error function.
> >
> > Given how central the robustness constraint is to the contributions of the paper, it feels like the paper should discuss the robust optimization objective more thoroughly and perhaps provide some notions or guarantees of what it is ultimately solving. If the error function is well-specified, then a large robustness set should push the selected $\rho$ to some suboptimal value.
> >
> > Conceptually, the interpolation aspect of the paper still feels incomplete to me. There are infinitely many ways to interpolate between two sampling policies, but this paper simply tests out three of these and settles on one based on a few computational experiments.

---

> > > ### Author Response · Authors · 2025-08-05
> > > **Further Response to Reviewer myd5**
> > >
> > > ​​Dear Reviewer,
> > >
> > > We appreciate your additional feedback. Below we address each point in sequence.
> > >
> > > ---
> > >
> > > > “Do you have a formal proof of this? I can believe that this does occur, but I don't immediately see it.”
> > >
> > > Below we provide a proof for why setting $\pi \propto \hat{e}$ leads to the trivial choice of $\rho$ when not using a robustness constraint.
> > >
> > > Starting from the variance estimate used in the optimization objective (line 131), by the Cauchy-Schwarz inequality, for any $\pi$ such that $\sum_{i=1}^{n} \pi(X_i) = n_b$ (i.e. satisfying the budget constraint), $\sum_{i=1}^{n} \frac{\hat{e}^2(X_i)}{\pi(X_i)} \geq \frac{(\sum_{i=1}^{n}\hat{e}(X_i))^2}{\sum_{i=1}^{n} \pi(X_i)} = \frac{(\sum_{i=1}^{n}\hat{e}(X_i))^2}{n_b} $. The equality holds when $\pi \propto \hat{e}$, which corresponds to $\pi^{(\rho)}$ with $\rho = 0$.
> > >
> > > We’ll add this proof in the revised version of the paper.
> > >
> > > ---
> > >
> > > > “It's a little confusing to see a justification for the robustness constraint arising from trying to avoid the trivial setting of $\rho$. The fact that this particular choice of $\pi$ results in $\rho=0$ is more of a reflection of the fact that this choice of $\pi$ is supposed to minimize a particular error function.”
> > >
> > > Our intention in incorporating a robustness constraint is not to force $\rho$ to be away from 0; rather, it is to underscore the critical importance of robustness, because simply tuning $\rho$ may result in poor sampling. In our view, the rationale for including robustness is clear and simple: tuning $\rho$ requires model error estimates, and we want to achieve good power when those are misspecified.
> > >
> > > ---
> > >
> > > > “Given how central the robustness constraint is to the contributions of the paper, it feels like the paper should discuss the robust optimization objective more thoroughly and perhaps provide some notions or guarantees of what it is ultimately solving. If the error function is well-specified, then a large robustness set should push the selected $\rho$ to some suboptimal value.”
> > >
> > > Good question—note that the magnitude of the robustness constraint is determined in a *data-driven* way; it is not hand-picked. We estimate the constraint value $c$ via cross-validation (see line 148). This in turn means that, if the errors are well-specified, the robustness set will *not* be large but will vanish. We will make sure to clarify this point.
> > >
> > > ---
> > >
> > > > “Conceptually, the interpolation aspect of the paper still feels incomplete to me. There are infinitely many ways to interpolate between two sampling policies, but this paper simply tests out three of these and settles on one based on a few computational experiments.”
> > >
> > > Our central assertion is that our method consistently produces an estimator that performs no worse than both the active and uniform estimators. The evidence presented in the paper strongly supports this claim; we made sure to test this claim across many data modalities, uncertainty scores, and inference problems. Exploring optimal paths is an interesting area for future research. Moreover, our framework’s flexibility—with multiple paths as well as robust formulations—should be seen as a strength rather than a weakness. Our stress-testing supports the geometric path + $\ell_2$ recommendation, yet we welcome the fact that alternative methods may ultimately outperform it.
> > >
> > > ---

---

### Official Review · Reviewer_mbvg · 2025-06-26

**Clarity:** 3
**Significance:** 2
**Originality:** 3
**Rating:** 4
**Confidence:** 3

**Summary:**

This paper proposes a robust sampling strategy for active statistical learning that guarantees performance no worse than uniform sampling. The core idea is to define a budget-preserving interpolation path between an arbitrary active sampling policy and uniform sampling, and to search along this path for a robust choice of the interpolation parameter $\rho$. The authors evaluate the method empirically on several datasets and demonstrate that it often outperforms both uniform sampling and the original active sampling policy.

While the conceptual contribution is interesting, the paper lacks clarity in several important algorithmic aspects and could benefit from a broader experimental evaluation. If these issues are addressed (especially clarification of optimization strategy and broader empirical baselines), the work would be a valuable contribution.

**Questions:**

**Weaknesses and Questions**

1. Algorithmic Clarity

   The process of obtaining $\rho_{\text{robust}}$ is not fully specified in Algorithm 1.

   - Do the authors perform grid search or use gradient-based methods to solve the minimax problem?
   - If grid search is used, how is the grid step chosen?
   - The paper should address the sensitivity of the solution to the grid step size or local minima, especially since small changes in $\rho$ can lead to large changes in the objective function.

2.  Misspecification Set $\mathcal{C}$

    The choice of the misspecification set $\mathcal{C}$ is not well justified.

     * The authors appear to assume an $\ell_2$-ball, but other choices (e.g., $\ell_\infty$, greedy perturbations, or adaptive constraints) could lead to different behaviors.
     * There is no discussion or comparison of how different choices of $\mathcal{C}$ affect robustness or optimization difficulty.
     * Given the known difficulty of minimax optimization (non-convexity, local optima), how do the authors ensure the optimal solution can be found?

3. Interpolation Path Design

   * In lines 112–113, the paper introduces linear and geometric interpolation paths between $\pi$ and $\pi_{unif}$.
   * However, the choice of these two is not motivated. Are there other potential interpolation strategies (e.g., entropy-based, adaptive weightings)?
   * A discussion of why only these two paths are chosen and their tradeoffs would strengthen the paper.

4. Experimental Coverage

   * The paper does not clearly specify which active sampling method $\pi$ is used in the experiments. This is essential for reproducibility and clarity.
   * Active learning encompasses many strategies (uncertainty sampling, margin sampling etc.). The current evaluation appears to be based on a single $\pi$.
   * To convincingly demonstrate the robustness of the proposed method, it would be helpful to evaluate it on multiple active sampling baselines.


5. Typos

   * line 76: "distribution of the unobserved labels" should be "observed labels".

**Ethical Concerns:**

["NO or VERY MINOR ethics concerns only"]

**Final Justification:**

The authors have addressed the major concerns raised in the initial review, particularly they give empirical results to support their theorem. I would like to raise my score to 4.

**Limitations:**

* The method relies on a well-estimated error $\hat{e}^2(\cdot)$. If this is poorly estimated (e.g., under small data regimes), the robust sampling policy might become unreliable.
* The minimax formulation adds significant optimization difficulty compared to standard active learning.
* The lack of theoretical guarantees about convergence or uniqueness of $\rho_{\text{robust}}$ is a concern.

**Paper Formatting Concerns:**

No format issue.

**Quality:**

3

**Strengths And Weaknesses:**

**Strengths**

* The motivation to robustify active sampling is strong, especially in settings where model misspecification or label noise can lead to degraded performance.
* The paper search over an interpolation path between uniform and active sampling for better estimators.
* Empirical results show improvements, supporting the theoretical analysis.
* The methodology is general and can potentially be applied to various existing active sampling strategies.

**Weakness**

Please refer to below questions section.

---

> ### Author Rebuttal · Authors · 2025-07-31
>
> Thank you for your comprehensive review and your valuable feedback. We detail our response below, and please kindly let us know whether our response addresses your concerns.
>
> ---
>
> > “The process of obtaining $\rho_{\mathrm{robust}}$ is not fully specified in Algorithm 1. Do the authors perform grid search or use gradient-based methods to solve the minimax problem? If grid search is used, how is the grid step chosen? The paper should address the sensitivity of the solution to the grid step size or local minima, especially since small changes in \rho can lead to large changes in the objective function.”
>
>
>
> We employed a grid search for $\rho$ in the outer loop, utilizing a step size of 0.01. The inner problem, being convex, is straightforward to solve and even possesses a closed-form solution when the constraint set is simple, such as the $\ell_2$-ball. If the objective function is Lipschitz, we can always locate an $\epsilon$-optimal point in terms of objective value, with the grid resolution determining $\epsilon$.The sensitivity of the results to the grid step size is a good question. We conducted new experiments to explore different step sizes and confirmed the robustness of our results to step-size selection, as shown below. Due to the restriction on uploading images/plots, the results are presented in a table format.
>
> On the Pew election dataset, our results show that a step size of 0.01 yielded $\rho_{\mathrm{robust}} = 0.43$, while a step size of 0.1 resulted in $\rho_{\mathrm{robust}} = 0.4$. The ESS gap between these two estimators is minimal, and both significantly outperform uniform and active baselines.
>
> | $n_b$  | Robust active (step_size=0.01) | Robust active (step_size=0.1) | Uniform | Active |
> |------|--------------------------------|-------------------------------|---------|--------|
> | 2794 | 2890                           | 2888                          | 2794    | 2802   |
> | 2875 | 2989                           | 2980                          | 2875    | 2894   |
> | 2956 | 3082                           | 3077                          | 2956    | 2951   |
> | 3037 | 3185                           | 3170                          | 3037    | 3060   |
> | 3118 | 3285                           | 3276                          | 3118    | 3104   |
> | 3199 | 3394                           | 3370                          | 3199    | 3192   |
> | 3280 | 3492                           | 3466                          | 3280    | 3257   |
>
>
>
> ---
>
> > “The choice of the misspecification set C is not well justified. The authors appear to assume an $\ell_2$-ball, but other choices (e.g., $\ell_{\infty}$, greedy perturbations, or adaptive constraints) could lead to different behaviors. There is no discussion or comparison of how different choices of $\mathcal{C}$ affect robustness or optimization difficulty. Given the known difficulty of minimax optimization (non-convexity, local optima), how do the authors ensure the optimal solution can be found?”
>
>
>
> As we briefly discussed in Section 2.2, we found that $\ell_2$ misspecification performs better than alternatives empirically. Further intuition is provided in the Appendix. For example, the $\ell_1$ norm is ineffective since it concentrates perturbations on a single error entry, leaving the remaining entries unchanged. Conversely, the $\ell_{\infty}$ norm uniformly distributes perturbations across all error entries, which is also not ideal.
>
> Regarding minimax optimization, as noted in the paper, the inner optimization becomes a convex problem—and thus easily solvable—when the misspecification set $\mathcal{C}$ is convex. This is crucial for the practicality of our proposal. The outer loop can then be solved via a simple, one-dimensional grid search over $\rho$.
>
>
> ---
>
>
> > “In lines 112–113, the paper introduces linear and geometric interpolation paths between $\pi$ and $\pi^{\mathrm{unif}}$. However, the choice of these two is not motivated. Are there other potential interpolation strategies (e.g., entropy-based, adaptive weightings)? A discussion of why only these two paths are chosen and their tradeoffs would strengthen the paper.”
>
>
>
> We have also included the Hellinger path in our paper, with its definition and validity detailed in Appendix B. Figure 4 illustrates a performance comparison between all three paths. These specific paths were chosen due to their simplicity. More generally, these are all examples of geodesic paths: paths that are roughly defined as “shortest paths.” Any geodesic path can be used in our proposal. We elaborate and give more examples of geodesics in Appendix B.
>
>
> ---
>
> > “The paper does not clearly specify which active sampling method $\pi$ is used in the experiments. This is essential for reproducibility and clarity. Active learning encompasses many strategies (uncertainty sampling, margin sampling etc.). The current evaluation appears to be based on a single $\pi$. To convincingly demonstrate the robustness of the proposed method, it would be helpful to evaluate it on multiple active sampling baselines.”
>
>
>
> We agree that stress testing the methodology with different sampling methods $\pi$ is important. We want to clarify that our paper *does* indeed test different $\pi$ and define these rules for each experiment. For instance, in Figure 2, $\pi(X) \propto \min \\{f(X), 1 - f(X)\\}$ (see line 222); in Figure 3, $\pi(X) \propto \hat{e}(X)$ (see line 233); in Figure 7, $\pi$ is based on verbalized LLM confidence (see line 273). We intentionally choose varying $\pi$ to demonstrate the robustness of the method.
>
> It is important to distinguish our framework from active learning despite their shared "active" nature, as their objectives are fundamentally different. Active inference aims to perform statistical inference on typically low-dimensional parameters. Consequently, many sampling strategies from active learning are not directly applicable or meaningful within the active inference framework, as it would be unclear how to perform valid statistical inference with those strategies. We will clarify these differences in the revised version.
>
> ---
>
>
> > “Typos. line 76: "distribution of the unobserved labels" should be "observed labels".”
>
>
>
> Thank you for raising this point. This is not a typo; rather, it highlights a crucial motivation behind our work. The true parameter $ \theta^* $ is influenced by the entire data distribution of $(X,Y)$. However, we a priori only observe unlabeled $X_i$ (line 73), no labels $Y_i$ (line 74).
>
> ---
>
> > “The method relies on a well-estimated error $\hat{e}^2$. If this is poorly estimated (e.g., under small data regimes), the robust sampling policy might become unreliable.”
>
>
>
> We want to emphasize that our robust sampling procedure remains effective even with a poor error estimation. A poorly estimated $\hat{e}$ would typically result in a deficient active estimator. However, with our additional robustness layer, a poorly estimated error leads to a larger constraint set $\mathcal{C}$, which in turn yields a larger $\rho_{\mathrm{robust}}$. This larger value pushes the sampling towards uniform sampling, ensuring the reliability of our method despite imprecise error estimation. Figure 6 illustrates the relationship between the interpolation coefficient and burn-in size: smaller burn-in sizes, leading to poorly estimated errors, result in larger values of $\rho_{\mathrm{robust}}$.
>
> Relatedly, we see that $\mathcal{C}$ is large when the burn-in size is small; see below. The corresponding dataset is the Pew election dataset, and $c$ is the magnitude of the $\ell_2$ constraint, i.e. $\mathcal{C} = \\{\epsilon: \\|\epsilon\\|\_2 \leq c\\}$.
>
> | Burn-in size  | $c$    |
> |------|------|
> | 606  | 4.43 |
> | 1213 | 2.78 |
> | 1819 | 1.95 |
> | 2426 | 0.91 |
> | 3639 | 0.49 |
>
> As expected, we observe a more conservative constraint when the errors are poorly estimated.
>
> ---
>
> > “The lack of theoretical guarantees about convergence or uniqueness of  $\rho_{robust}$ is a concern.”
>
>
> To reiterate our previous point about computational efficiency: when the constraint set $\mathcal{C}$ is convex, the inner problem of the minimax formulation becomes a convex problem. This simplifies optimization and prevents convergence to local minima. The effect of performing a one-dimensional grid search over [0,1] to search over $\rho$ is minimal, as illustrated above.
>
> Furthermore, in line 767 in the Appendix, we offer a theoretical guarantee: $\hat{\rho}$ will converge to $ \rho^* $ provided $ \rho^* $ is unique and error consistency is achieved. If a consistent error estimator is available, the misspecification set will asymptotically vanish. Consequently, the consistency of $\rho_{\mathrm{robust}}$, the asymptotic normality of its corresponding estimator w.r.t $\rho_{\mathrm{robust}}$, and the property of its asymptotic variance can be derived by following the same steps used in the proofs of Theorem 1 and Theorem 2. However, formally working out these details is beyond the scope of the present work.
>
> ---
>
> [1] Zrnic, Tijana, and Emmanuel Candes. "Active Statistical Inference." International Conference on Machine Learning. PMLR, 2024.

---

> > ### Comment · Reviewer_mbvg · 2025-08-04
> >
> > Thank you for the author's response. Most of my concerns have been addressed, and I would like to revise my score.

---

> > > ### Author Response · Authors · 2025-08-05
> > >
> > > Thank you for taking the time to read our response and for thoughtfully reconsidering your evaluation! We greatly appreciate your careful review and constructive feedback.

---

### Official Review · Reviewer_svoR · 2025-06-29

**Clarity:** 3
**Significance:** 3
**Originality:** 3
**Rating:** 4
**Confidence:** 3

**Summary:**

This paper proposes a methodology for performing appropriate labeling in situations where labeling each instance is expensive. While classical active learning methods adaptively determine where to perform labeling and place emphasis on prediction, this paper places emphasis on statistical inference of the model associated with data observation. Existing active statistical inference based on sampling from a random distribution suffers from the problem that when the error value is large, i.e., the sampling probability is small, the results are worse than uniform sampling. In contrast, this paper proposes a robust sampling rule that lies between uniform sampling and active statistical inference, bridging the two. Inference based on the proposed method has been shown to have a smaller asymptotic variance than that of uniform sampling and active statistical inference under appropriate assumptions. Additionally, numerical experiments have been conducted to compare the proposed method with uniform sampling and active statistical inference.

**Questions:**

-Questions

Please clarify the following unclear points. If these unclear points are appropriately revised, I will raise the score.

1. Since the algorithm randomly decides whether to observe each instance, it is possible that all instances are observed in extreme cases. Does this paper allow the budget $n_b$ to be exceeded? If so, can the same claims as in this paper be made while keeping the total number of observations within $n_b$ by adaptively changing the sampling distribution for each observation, similar to active testing (Kossen et al., 2021)?

2. In lines 9, 101, and 297, is this claim asymptotic? Does it hold for finite samples?

3. The explanation of the inconsistency in the axis scales in all figures, including Figure 1, and how to read the figures should be clarified.

4. References to related research. The differences between active testing and the present setting should be mentioned. In addition, the issues with performing statistical inference after classical active learning should be mentioned. My understanding is that the bias caused by not randomly selecting instances is the problem. Is this correct? If so, related research on statistical inference after decision-making based on statistical models, such as post-selection inference (Lee et al., 2016), should be mentioned to see if the bias of active selection can be corrected. Perhaps this is difficult because the selection of instances by active learning does not fit into the framework of post-selection inference. Is that correct?

Lee et al., Exact post-selection inference, with application to the lasso, The Annals of Statistics 2016


5. It is assumed that $\hat{e} (\cdot)$ converges in probability to $e (\cdot)$, but I am not sure if this assumption is strong. Are sufficient conditions given?


-Comments

The following points are not necessarily required to be addressed, but they caught my attention.

1. Is $y$ limited to 0 or 1? Can this be extended to general distributions?

2. If $\mathcal{C}$ is an $l_2$-ball and the radius is reduced to 0 (as $n \to \infty$), can the same theorem be derived?

3. I think it would be better to clarify what the expected values refer to, such as line 85 and Equation (2). I think that expected values of $X$ and those of $(X, Y)$ are mixed together.

**Ethical Concerns:**

["NO or VERY MINOR ethics concerns only"]

**Final Justification:**

The authors' rebuttal appropriately addressed the criteria for raising the score that I presented in Questions, particularly the resolution of ambiguities and the relevance to existing research. In my opinion, this paper has no clear flaws that would warrant rejection, so I have increased my score to 4. However, I believe that this paper can be improved regarding exceeding the budget $n_b$. In the problem setting, we control the number of observations in expectation, so exceeding the budget $n_b$ is acceptable. While I think this is unavoidable because of designing a sampling distribution, I think it should also be strictly within the budget $n_b$.

**Limitations:**

Yes. However, as I understand it, it seems that exceeding the budget is permitted, so the possibility that strict budget constraints may not be met should be added to the limitations. On the other hand, this study proposes a new  method for robust statistical inference in active learning, and therefore does not fall under potential negative societal impact.

**Quality:**

3

**Strengths And Weaknesses:**

The strengths of this paper are as follows:

1. It solves the problem of existing active statistical inference, which is that it performs worse than uniform sampling when the error is large, by appropriately linking the two.
2. It extends to general M-estimators. It also extends to robust estimation by using the admissible set $\mathcal{C}$ to allow for the effects of misidentification that arise when estimating the error.

On the other hand, the weaknesses of this paper are as follows:

1. Insufficient reference to related research. The relationship and differences with active testing (Kossen et al., 2021) should also be mentioned. In addition, the problems of statistical inference when using conventional active learning (e.g., selecting from the most uncertain points) should also be mentioned.
2. Lack of clarity in the problem setting and experimental results. If my understanding is correct, the proposed method seems to allow exceeding the prior budget. Furthermore, the explanation of the strength of the assumption that the estimate $\hat{e} (\cdot)$ in the algorithm converges in probability to the error $e (\cdot)$ is insufficient. In addition, the intervals between the numerical values on the axes of the figures in the numerical experiments are not consistent, making it difficult to determine what these figures represent. As a result, the effectiveness of the proposed method cannot be correctly confirmed by numerical experiments.

[Quality: 3]
The proposed method provides a theory on the asymptotic variance of estimators using an algorithm and extends it to general M estimators. Furthermore, it proposes a robust method for the effect of misidentification of the error function. Additionally, it mentions the issues of the proposed method, such as the selection of $\mathcal{C}$ and its impact.

[Clarity: 2 $\to$ 3]
There are several unclear points regarding the proposed method. First, it is unclear whether the algorithm allows the budget to be exceeded. Second, the explanation of the strictness of the error estimation, i.e., that $\hat{e} (\cdot)$ converges in probability to $e (\cdot)$, is insufficient. Third, the intervals between the axis scales in each figure are not consistent, making it difficult to interpret the figures and, consequently, to determine whether the numerical experimental results are significant.
* After the rebuttal, the authors addressed all three points of ambiguity, so I raised Clarity to 3.

[Significance: 3]
The proposed method offers a new approach to the problem of existing methods, where the results become worse than uniform sampling when the error function is large, and provides several extensions. This contributes new insights to the field of statistical inference for active learning. Additionally, the paper points out that there is room for further research on the selection of $\mathcal{C}$ and its impact on robustness.

[Originality: 2 $\to$ 3]
The proposed method suggests the possibility of designing a new distribution that connects uniform sampling and active statistical inference, and is original in that it also considers the impact of misidentification in estimation. However, it lacks comparison with several important related studies. For example, active testing (Kossen et al., 2021) also focuses on statistical inference in active learning rather than prediction, and like this paper, it designs a distribution proportional to the error in each instance and makes decisions based on the realized values from that distribution. It should also mention the problems of using conventional active learning for statistical inference. In the current draft, it seems preferable to perform classical active learning, select samples that minimize prediction, and then perform statistical inference on the resulting model.
* After the rebuttal, the authors clearly explained the relevance of their study to existing studies, so I raised Originality to 3.

Based on the above, I rated the paper as 3 due to room for improvement in clarity and originality.

Kossen et al., Active Testing: Sample–Efficient Model Evaluation, ICML 2021

---

> ### Author Rebuttal · Authors · 2025-07-31
>
> We appreciate the reviewer's comprehensive summary and helpful feedback. We have addressed each of your points below. Please let us know if these responses adequately address your suggestions.
>
> ---
>
> > “Since the algorithm randomly decides whether to observe each instance, it is possible that all instances are observed in extreme cases. Does this paper allow the budget  to be exceeded? If so, can the same claims as in this paper be made while keeping the total number of observations within by adaptively changing the sampling distribution for each observation, similar to active testing (Kossen et al., 2021)?”
>
>
> Like prior work [1], our algorithm enforces the budget *in expectation*. However, since the number of labeled data points is a sum of independent Bernoullis, a standard Hoeffding argument guarantees that the realized labeling rate will closely match the budget with high probability. Specifically, the labeling ratio will not exceed $\frac{n_b}{n} + \epsilon$ with a probability of $1 - \delta$, provided that $n > \frac{\log (1/\delta)}{2 \epsilon ^2}$ for any $\epsilon, \delta > 0$. (Alternatively, this can be interpreted as: if we run the algorithm with a slightly reduced budget $n_b - \epsilon$, then with high probability the budget $n_b$ will be met.)
>
> Active testing [2] pursues a different objective (high‑precision risk estimation for a fixed model) and uses a distinct estimator so the program is very different. That said, the active data collection is closely related to the setting of active inference. Thank you for pointing this out–we will add a discussion of the connection between active inference and active testing in the revision. However, our primary contribution lies in *robustifying* the sampling process, rather than proposing a new active approach to statistical inference. We believe that this is a novel contribution.
>
> ---
>
> > “In lines 9, 101, and 297, is this claim asymptotic? Does it hold for finite samples?“
>
>
> Yes, our theoretical claim is asymptotic, as supported by Theorems 1 and 2. We will make sure to clarify this. Our empirical findings in the experimental section show this claim holds true in finite samples.
>
>
> ---
>
>  > “The explanation of the inconsistency in the axis scales in all figures, including Figure 1, and how to read the figures should be clarified.”
>
>
> Thank you for flagging this issue. We acknowledge the presence of multiple axis scales within the paper, specifically for sample size, coverage, and interpolation coefficients.
>
> For coverage and interpolation coefficients, the scale remains consistent across all figures, ranging from 0 to 1. The scale for the budget $n_b$ varies between figures due to several factors, including dataset size (for example, we set the budget as a percentage of the overall dataset size, e.g. 10%) and burn-in size (which is included in the budget). The y-axis, representing the effective sample size (ESS), is determined by how much the procedure gains over uniform sampling, which is in turn determined by the accuracy of the predictions and uncertainties.
>
> To accurately interpret the figures, the reader should understand the concepts of *effective sample size* (ESS) and *coverage*.
>
> ESS quantifies a method’s efficiency compared to uniform sampling. For instance, if our method utilizes a labeling budget of $n_b = 500$ and achieves an ESS of $n_{\mathrm{eff}} = 600$, this means that our method with 500 labels is as effective as uniform sampling with 600 labels. A higher ESS relative to the baseline signifies greater efficiency. (The uniform estimator by definition will always have an ESS equal to its budget ($n_{\text{eff}} = n_b$)). Readers can also refer to the definition of effective sample size provided in Lines 193-200 of Section 4.
>
> Coverage captures how often the constructed confidence intervals cover the quantity of interest. Coverage should be at least 0.9 for the error to be controlled, and ideally it should be as close to 0.9 as possible.
>
> ---
>
> > “References to related research. The differences between active testing and the present setting should be mentioned. In addition, the issues with performing statistical inference after classical active learning should be mentioned. My understanding is that the bias caused by not randomly selecting instances is the problem. Is this correct? If so, related research on statistical inference after decision-making based on statistical models, such as post-selection inference (Lee et al., 2016), should be mentioned to see if the bias of active selection can be corrected. Perhaps this is difficult because the selection of instances by active learning does not fit into the framework of post-selection inference. Is that correct?”
>
>
>
> Thank you for this suggestion. We will expand on the distinctions between active testing and active inference in the revised version, as discussed above. However, we want to reiterate: our primary contribution is to *robustify* the sampling of active inference. This means our sampling method performs well regardless of the uncertainty scores obtained, a benefit not offered by active testing. It is true that classical active learning's greedy label selection can disrupt inference by introducing bias. Our method, consistent with [1], effectively mitigates this bias using an inverse probability weighting method.
>
> Post-selection inference (POSI) [3] addresses a fundamentally different problem: adaptively selecting hypotheses based on *fixed* data. Our framework, however, focuses on adaptively choosing data based on *fixed* hypotheses. These represent two different sources of bias, each with its own unique challenges. The bias mitigation techniques developed and leveraged in the two lines of work are entirely distinct.
>
> ---
>
> > “It is assumed that $\hat{e}$ converges in probability to $e$, but I am not sure if this assumption is strong. Are sufficient conditions given?”
>
>
> Thank you for your question. The assumption that $\hat{e}$ converges to $e$ follows from classical arguments. As the size of the training or burn-in dataset increases, more information becomes available to estimate the conditional expectation. Flexible estimators, such as the multilayer perceptron (MLP) and XGBoost models used in the paper's experiments, are universal approximators. With sufficient data, they can learn complex relationships and provide a consistent estimate of the true conditional expectation, $e$, assuming the model is correctly specified and $e$ belongs to a sufficiently smooth or low-complexity class. While the rate of convergence depends on the specific estimator and the complexity of the true function, the convergence itself is a standard result from learning theory.
>
> (More generally, this type of assumption is common in the field of semiparametric inference. For instance, the widely-used doubly robust estimator [4, 5], which is closely related to our estimator, relies on consistent estimation of nuisance functions.)
>
> ---
>
> > “Is $Y$ limited to 0 or 1? Can this be extended to general distributions?”
>
>
> Our methodology does not assume $Y$ to be restricted to 0/1. Indeed, we consider general convex M-estimation problems, such as linear regression. For example, in the experiment on US Census data in Figure 5, $Y$ corresponds to income and is thus a continuous variable.
>
>
> ---
>
> > “If $\mathcal{C}$ is an $\ell_2$-ball and the radius is reduced to 0 (as $n \rightarrow \infty$), can the same theorem be derived?”
>
>
>
> This is a good question. Roughly speaking, if the estimated error is consistent and the constraint set $\mathcal{C}$ vanishes as $n \rightarrow \infty$, the consistency of $\rho_{\mathrm{robust}}$, the asymptotic normality of its corresponding estimator w.r.t $\rho_{\mathrm{robust}}$, and the property of its asymptotic variance can be derived by following the same steps used in the proofs of Theorem 1 and Theorem 2. This supports the asymptotic validity of the robust sampling procedure. However, formalizing this precisely is beyond the scope of the present work.
>
> ---
>
> > “I think it would be better to clarify what the expected values refer to, such as line 85 and Equation (2). I think that expected values of $X$ and those of $(X,Y)$ are mixed together.”
>
>
>
> The expectation is consistently defined throughout the paper: all expectations $\mathbb{E}$ average out *all* randomness in the argument, unless a conditional expectation is used, in which case the conditional expectation corresponds to a random variable that is measurable with respect to the variable being conditioned on.
>
> ---
>
> [1] Zrnic, Tijana, and Emmanuel Candes. "Active Statistical Inference." International Conference on Machine Learning. PMLR, 2024.
>
> [2] ​​Kossen, Jannik, et al. "Active testing: Sample-efficient model evaluation." International Conference on Machine Learning. PMLR, 2021.
>
> [3] Lee, Jason D., et al. "Exact post-selection inference, with application to the lasso." (2016): 907-927.
>
> [4] Glynn, Adam N., and Kevin M. Quinn. "An introduction to the augmented inverse propensity weighted estimator." Political analysis 18.1 (2010): 36-56.
>
> [5] Robins, James M., Andrea Rotnitzky, and Lue Ping Zhao. "Estimation of regression coefficients when some regressors are not always observed." Journal of the American statistical Association 89.427 (1994): 846-866.

---

> > ### Comment · Reviewer_svoR · 2025-08-01
> >
> > Thank you for your response. All of my questions except for question 3 have been resolved. However, question 3 remains unclear.
> >
> > My question concerns the fact that the intervals between the numbers on the axis in the figure are not equal. For example, in the left figure of Figure 1, the four numbers on the horizontal axis—1291, 1471, 1677, and 1912—are evenly spaced, but the differences between these numbers are not equal: $1471-1291=180$, $1677-1471=206$, $1912-1677=235$. Similarly, the vertical axis has seven numbers evenly spaced, but the differences between these numbers are not equal. And these results are connected by three lines, but how should we interpret these lines in a graph where the intervals between the numbers on the horizontal and vertical axes do not match? The same phenomenon occurs in Figure 2 and subsequent figures. I believe I may have misunderstood how to read the figures, so I would appreciate an explanation of how to interpret these figures, taking into account the reason why the intervals between the numbers on the axes do not equal.

---

> > > ### Author Response · Authors · 2025-08-02
> > > **Reply to Reviewer svoR regarding Question 3**
> > >
> > > Dear Reviewer,
> > >
> > > Thank you for your thoughtful feedback. We are glad that we have addressed most of your concerns. In all of our plots, we employ logarithmic scaling. Because the interval width—and consequently the ESS—scales log-linearly with sample size, a log scale produces an approximately linear curve that is much clearer to interpret. We will include this explanation in the revised manuscript. Thank you again for your valuable comments!

---

> > > > ### Comment · Reviewer_svoR · 2025-08-02
> > > >
> > > > Thank you for explaining the interpretation of the figure. My misunderstanding has been cleared up. Therefore, since the unclear points I mentioned in Questions have been resolved, I will raise my initial score to 4.

---

> > > > > ### Author Response · Authors · 2025-08-02
> > > > > **Thank you**
> > > > >
> > > > > Thank you for your feedback! We’re delighted to have addressed all your questions and appreciate you updating your score. If you have any other questions, please don’t hesitate to let us know. Thanks again!

---

### Official Review · Reviewer_hhQ5 · 2025-07-01

**Clarity:** 4
**Significance:** 2
**Originality:** 3
**Rating:** 5
**Confidence:** 4

**Summary:**

In this paper, the authors propose a sampling scheme for statistical inference that is a mixture of uniform and uncertainty based sampling schemes. The authors note that the uncertainty based sampling depends on the quality of the model’s uncertainty estimates, and can even perform worse than random sampling when these estimates are poor. Subsequently, they propose a robust scheme that guarantees the performance would be bounded by that of uniform sampling. Results seem to indicate that the proposed scheme performs well with a range of qualities of uncertainty estimates.

**Questions:**

See strengths and weaknesses

**Ethical Concerns:**

["NO or VERY MINOR ethics concerns only"]

**Final Justification:**

I want to thank the authors for addressing all my questions and running additional experiments to address issues I pointed out as weaknesses. With this additional context, I am raising my score from a "borderline accept" to "accept".

**Limitations:**

Yes

**Quality:**

3

**Strengths And Weaknesses:**

The central idea of using error function estimates along a path connecting the two sampling schemes is simple and elegant. The main theorems are clearly stated and the paper does a good job of building intuition first with a concrete example (mean estimation in section 2). I had a few concerns and observations but I am somewhat inclined for the paper to be accepted:

1. I am not entirely convinced how practical this method might be given its reliance on some hand-picked hyperparameters, namely the set of misspecification (C) and the choice of path. While the choices made in the paper seem to work well, they do not sufficiently convince me how the results might generalize.
2. Relatedly, in Fig 3, I am not sure how much of the performance difference can be attributed to the overall scheme vs. just the choice of path. Just eyeballing Fig 4 against Fig 3, it appears the choice of path leads to a substantial difference in performance, with a similar magnitude to the performance difference in Fig 3. It would be useful to see where, say, a linear path lands in Fig 3. Essentially some more ablation studies to really understand the benefits of the proposal.
3. For the choice of C: the l_2 norm constraint doesn’t seem very intuitive. What if the error estimates are off systematically? We might suspect such systematic differences for the regions where the model is uncertain vs. where it is fairly certain. Would this still bound the overall variance? Would it likely just converge to uniform sampling in practice?
4. Minor: in Fig 2, consider using the same scaling for both figures so it’s easy to quickly compare the two. You could even combine the two figures and just use different formatting for the curves (-- vs .-). In line 214, “we collect all labels Y_i”, I presume you mean just the burn-in labels. In Fig 3, curious why are we using such odd burn-in sizes?

---

> ### Author Rebuttal · Authors · 2025-07-31
>
> Thank you for your thorough and constructive feedback. See below for our responses to your points, and please let us know if these responses adequately address your suggestions.
>
> ---
>
> >“I am not entirely convinced how practical this method might be given its reliance on some hand-picked hyperparameters, namely the set of misspecification ($\mathcal{C}$) and the choice of path. While the choices made in the paper seem to work well, they do not sufficiently convince me how the results might generalize.”
>
>
> We appreciate this comment. We have tried several options for both the path and the misspecification set, and across many datasets (including varying data modalities) and uncertainty measures. We have found that the method is reliably never worse than uniform sampling, and often outperforms uniform sampling and active inference by a substantial margin. We have done our best to stress test the methodology; if you have suggestions for additional experiments for validating these observations, we will gladly run them.
>
> We would also argue that two parameters (misspecification set and choice of path) is not an impractically large number of user-chosen parameters. In our extensive evaluations we have found the $\ell_2$-constraint and geometric path to be simple and lead to good performance. This is why, to maximize practicality and ensure running the method is as streamlined as possible, we recommend them as a default.
>
> ---
>
>
> > “Relatedly, in Fig 3, I am not sure how much of the performance difference can be attributed to the overall scheme vs. just the choice of path. Just eyeballing Fig 4 against Fig 3, it appears the choice of path leads to a substantial difference in performance, with a similar magnitude to the performance difference in Fig 3. It would be useful to see where, say, a linear path lands in Fig 3. Essentially some more ablation studies to really understand the benefits of the proposal.”
>
>
> Thank you for this insightful suggestion. Our core claim is that our procedure consistently yields an estimator that performs no worse than either the active or uniform estimator. However, we acknowledge that the degree to which it outperforms these baselines is highly variable, depending on the specific datasets, paths, and models used.
>
> To address your question, we conducted new experiments comparing the geometric path, linear path, and the baselines (active and uniform) in the setting of Figure 3. The results, presented below, demonstrate that while the linear path does not achieve the same performance as the geometric path, it consistently performs no worse than either of the two baselines. This finding supports the above claim. We will add these new results to the revised version of the paper. Due to the restriction on uploading images/plots, the results are presented in a table format.
>
> Burn-in size 606:
>
> | $n_b$ | Robust active (geometric) | Robust active (linear) | Active | Uniform |
> |-------|---------------------------|------------------------|--------|--------|
> |  1396    |            1511               |             1434         |     1195   |   1396    |
> | 1518    |                 1652          |              1569         |     1264   |     1518   |
> | 1760    |                  1932         |              1836         |     1422   |     1760   |
> | 2003   |                    2205       |               2097         |    1652   |     2003   |
>
> Burn-in size 1213:
>
> | $n_b$ | Robust active (geometric) | Robust active (linear) | Active | Uniform |
> |-------|---------------------------|------------------------|--------|--------|
> |  1807    |            1896               |          1844            |     1744   |   1807    |
> | 2023    |                 2142         |               2073        |     1954   |     2023   |
> | 2239    |                  2393         |               2286        |     2058   |     2239   |
> | 2455   |                    2641       |             2519           |    2218   |     2455   |
>
> Burn-in size 1819:
>
> | $n_b$ | Robust active (geometric) | Robust active (linear) | Active | Uniform |
> |-------|---------------------------|------------------------|--------|--------|
> |  2340    |            2418               |            2358           |     2288   |   2340    |
> | 2529    |                 2620          |               2553         |     2481   |     2529   |
> | 2718    |                  2854         |                2765        |     2646   |     2718   |
> | 2907   |                   3066        |             2963           |     2823   |     2907   |
>
> Burn-in size 2426:
>
> | $n_b$ | Robust active (geometric) | Robust active (linear) | Active | Uniform |
> |-------|---------------------------|------------------------|--------|--------|
> |  2793    |            2866               |         2840          |     2796   |   2793    |
> | 2955   |               3064          |                3033       |     2982   |     2955   |
> | 3117    |               3265         |              3189        |     3131   |     3117   |
> | 3279   |                3458        |             3364        |     3287  |     3279   |
>
> Burn-in size 3639:
>
> | $n_b$ | Robust active (geometric) | Robust active (linear) | Active | Uniform |
> |-------|---------------------------|------------------------|--------|--------|
> |  3881    |            3963               |           3938        |     3923   |   3881    |
> | 3935   |               4024         |              4022         |     4010   |     3935   |
> | 4043    |               4166         |               4147       |     4126   |     4043   |
> | 4151   |                4308        |               4285      |     4251  |     4151  |
>
>
>
> ---
>
> > “For the choice of $\mathcal{C}$: the $\ell_2$ norm constraint doesn’t seem very intuitive. What if the error estimates are off systematically? We might suspect such systematic differences for the regions where the model is uncertain vs. where it is fairly certain. Would this still bound the overall variance? Would it likely just converge to uniform sampling in practice?”
>
>
> We agree that a simple $\ell_2$ norm constraint may not always be the most powerful choice of $\mathcal{C}$. Zooming out, our method can in principle be combined with *any* choice of $\mathcal{C}$, including one where we learn regions of the space where scores are systematically overconfident or underconfident. At a high level, our method (1) learns $\mathcal{C}$ (in our experiment, the “learning” is a simple fitting of $c$ through cross-validation), and (2) solves a robust optimization problem with $\mathcal{C}$ in place. Your suggestion is an interesting choice of step (1).
>
> We developed a dataset featuring a central "hard" region ($|X| \leq 2$) flanked by two "easy" regions ($2<|X|<5$). In the easy regions, error data was sampled from $\mathcal{N}(2, 0.05)$. In the hard region, error was drawn from $\mathcal{N}(1, 0.25)$. The estimator of error, $\hat{e}(X)$, is designed to underestimate the error in the hard region and overestimate the error in the easy region. Specifically, $\hat{e}(X) = 0.5$ for $|X| \leq 2$, and $\hat{e}(X)=2.5$ otherwise.
>
> Subsequently, we trained a meta-classifier, a gradient boost classifier, $h(X)$, to identify these regions solely based on the performance of $\hat{e}(X)$, without prior knowledge of the region boundaries.
>
> This approach proved highly effective, with the meta-classifier achieving over 99% accuracy in identifying the regions. This demonstrates our success in learning the error regions and enables us to separate the constraint set $\mathcal{C}$ based on these distinctions. For instance, $\mathcal{C}$ can be defined as $ \\| \epsilon_{\mathrm{easy}} \\|\_2 \leq c_{\mathrm{easy}} $ for the easy region ($2<|X|<5$) and $\\|\epsilon_{\mathrm{hard}}\\|\_2 \leq c_{\mathrm{hard}} $ for the hard region ($|X| \leq 2$). Or even simpler, we can only optimize over hard regions, i.e. $c_{\mathrm{easy}}=0$. While these regions' dimensions are not fixed and depend on $X$, this presents no practical difficulties because we have complete information about $X$.
>
>
> Next, we compared this structured constraint with the global constraint. Here, for the structured constraint, we only optimize over the hard region. The following table shows the result when $n=7000$, $n_b=1400$, and $\pi \propto \hat{e}$.
> | Method | ESS | ESS Gain (%) |
> |-------|---------------------------|------------------------|
> | Uniform | 1400 | 0.00% |
> | Active | 1213 | -13.3% |
> | Robust active (global) | 1491 | 6.5% |
> | Robust active (structured) | 1495 | 6.8% |
>
> We found that incorporating the structured constraint provided a slight gain in ESS over the global constraint while reducing the constraint size ($c_{\mathrm{global}} = 85$ vs. $c_{\mathrm{hard}} = 75$). This suggests that a more focused perturbation can be beneficial when we have strong knowledge of confident regions. However, we note that the global constraint remains a simple and practical approach given the limited gain.
>
>
>
>
> ---
>
> > “Minor: in Fig 2, consider using the same scaling for both figures so it’s easy to quickly compare the two. You could even combine the two figures and just use different formatting for the curves (-- vs .-). In line 214, “we collect all labels $Y_i$”, I presume you mean just the burn-in labels. In Fig 3, curious why are we using such odd burn-in sizes?”
>
>
> This is a helpful suggestion. We have revised Figure 2 as suggested, aligning the x-axes. Yes, "we collect all labels $Y\_i$" refers to the burn-in labels; we will rephrase this sentence for clarity. The burn-in sizes in Figure 3 are set by the proportion of the burn-in data to the whole data (e.g., 10% for a burn-in size of 606), which explains their seemingly unusual appearance. We will clarify this in the revised version.
>
> ---

---

> > ### Comment · Reviewer_hhQ5 · 2025-08-05
> >
> > I want to thank the authors for addressing all my questions and running additional experiments to address issues I pointed out as weaknesses. With this additional context, I am raising my score from a "borderline accept" to "accept".

---

> > > ### Author Response · Authors · 2025-08-06
> > > **Thank you**
> > >
> > > Thank you for reviewing our response and thoughtfully reconsidering your evaluation! We’re delighted to have addressed all your questions and truly appreciate your updated score. Your thorough feedback has been invaluable in strengthening our paper.

---

### Note · Authors · 2025-08-14

With the conclusion of the discussion period, we would like to extend our gratitude to all reviewers and the AC for their insightful feedback and constructive guidance, which have significantly strengthened our paper. We are delighted that we addressed reviewer concerns, leading to score increases from hhQ5, svoR, and mbvg.

The discussion solidified the core contributions of our work: a robust, practical sampling framework that is guaranteed to perform no worse than uniform or active sampling, and often substantially better. We clarified the novelty of our work in relation to prior works (per svoR, myd5), and added new experiments demonstrating:

- Our method is robust to hyperparameter choices, with the recommended geometric path a simple and strong default (per hhQ5, usPt).

- The framework remains effective when error estimates are systematically off (per hhQ5).

-  Our method is robust to step sizes and doesn’t rely on a well-estimated error (per mbvg).

-  Robust optimization yields gains across different choices of $\pi$ (per myd5).

- Standard deviation of effective sample size reflects stability in the results of multiple trial runs (per myd5).

We will incorporate all clarifications, additional experimental results, and discussions into the revised manuscript.

---

### Decision · Program_Chairs · 2025-09-17

**Decision:**

Accept (poster)

**Comment:**

This paper studies a method for sampling data in order to achieve more data efficient statistical estimation. The reviewers appreciated the clear description, the simplicity, elegance, and generality of the proposed method, and good empirical results. However, the reviewers also noted a lack of rigorous analysis (both theoretical and experimental) of error bars, hyperparameter choices, and methodological choices which were mostly addressed during the rebuttal period. Additionally, some aspects of the exposition were unclear such as (1) the (expected) budget constraint, (2) relationship to prior work, and (3) burn-in period of the algorithm (not in the main algorithm box of Algorithm 1), though were clarified during the rebuttal period. Overall, I recommend acceptance, though weakly given the large number of changes from the submission. If this paper is accepted, we expect (prior to camera-ready) the authors to take reviewer recommendations for revision of the paper seriously and integrate all experiments and clarifications mentioned by the authors in the rebuttal.